# Alleviating Observation Bias via Causal-Invariant Meta-Learning for Unbalanced Incomplete Multi-view Clustering

**Jiaqi Jin** [1]  **Siwei Wang** [2]  **Taichun Zhou** [1]  **Zhibin Dong** [1]  **Siqi Wang** [1]  **Miaomiao Li** [3]  **Xinwang Liu** [1]  **En Zhu** [1]

## Abstract

In real-world scenarios, multi-view data often exhibits significant imbalance in missing patterns across views, where observation rates vary substantially among different views. Such observation bias makes it difficult for cross-view associations learned from limited complete samples to generalize to incomplete samples, leading to challenging cross-view recovery. Meanwhile, observation bias acts as a confounder, causing clustering predictions to spuriously depend on low-missing-rate views. To address these challenges, we propose CIMLN, a novel **C**ausal-**I**nvariant **M**eta-**L**earning **N**etwork that alleviates observation bias for unbalanced incomplete multi-view clustering. The context-aware meta-generation module formulates view recovery as a meta-learning task, enabling rapid adaptation to incomplete samples by encoding global statistical relationships through context information. The causal-invariant structure learning module constructs counterfactual scenarios by artificially masking low-missing-rate views, enforcing clustering consistency across different observation patterns. Extensive experiments on eight benchmarks demonstrate the effectiveness of CIMLN. The code is available at https://github.com/jinjiaqi1998/CIMLN.

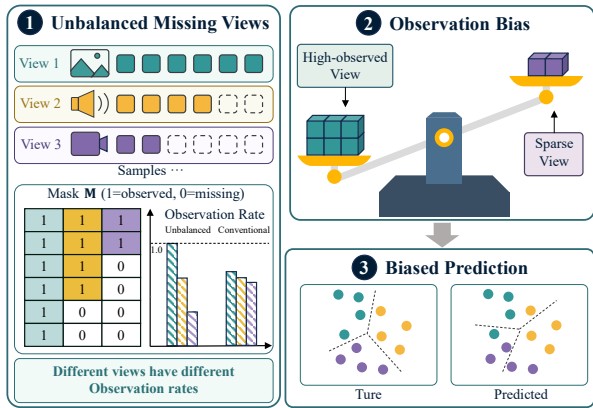

*Figure 1.* Illustration of the unbalanced missing pattern in multi-view data. The first section demonstrates this unbalanced missing scenario using three views as an example, employing a bar chart to compare the observation rates of each view under both "unbalanced" and "conventional" settings. The second and third sections illustrate the observation bias induced by this unbalanced missing pattern, as well as the subsequent problem of biased prediction.

## 1. Introduction

The widely recognized premise that leveraging more data sources improves learning performance has driven the rapid development of multi-view learning (Ren et al., 2024; Cui et al., 2025; Chen et al., 2025a; Zhou et al., 2026; Guan et al., 2025b;a). By characterizing the same object from multiple heterogeneous sources, multi-view data provides more comprehensive and complementary information than any single view alone. Multi-view clustering (Chen et al., 2025b; Xu et al., 2025; Yu et al., 2025; 2024) implicitly assumes that all views are fully observable and aims to integrate information across views to discover the inherent grouping structure of data in an unsupervised manner. However, in real-world applications, factors such as high acquisition costs and privacy restrictions often render multi-view data only partially available, giving rise to the problem of Incomplete Multi-View Clustering (IMVC) (Wen et al., 2025; Chen et al., 2024; Zhao et al., 2025).

Both recovery-based (Wang et al., 2025c; Liu et al., 2023) and recovery-free IMVC methods (Wang et al., 2020; Feng et al., 2024; Xue et al., 2024) primarily focus on scenarios where views are randomly or uniformly missing. However, in a broader range of practical applications, missing patterns often exhibit significant imbalance, i.e., the observation rates vary considerably across different views, as illustrated in Figure 1. In medical diagnosis, medical imaging examinations (Cui et al., 2026) typically have observation

---

[1]College of Computer Science and Technology, National University of Defense Technology, Changsha, China [2]Intelligent Game and Decision Lab, Academy of Military Science, Beijing, China [3]School of Computer Science and Engineering, Changsha University, Changsha, China. Correspondence to: Siwei Wang <wangsiwei13@nudt.edu.cn>, Xinwang Liu <xinwangliu@nudt.edu.cn>, En Zhu <enzhu@nudt.edu.cn>.

*Proceedings of the 43rd International Conference on Machine Learning*, Seoul, South Korea. PMLR 306, 2026. Copyright 2026 by the author(s).

rates below 20% due to high costs, whereas routine blood tests approach 100%. In financial credit assessment, emerging e-commerce behavioral data has an availability rate of only 15-30% due to user privacy protection and platform fragmentation. In remote sensing, optical imagery suffers from cloud occlusion with observation rates around 40%, while SAR radar data achieves over 80%. Clustering on datasets with such significant disparities in view availability is termed Unbalanced Incomplete Multi-View Clustering (UIMVC).

Current IMVC methods are primarily designed for relatively balanced missing scenarios (Chowdhury et al., 2025; Liu et al., 2024; Long et al., 2023). Under unbalanced missing patterns, however, the information imbalance across views introduces severe observation bias, posing significant challenges. On one hand, observation bias forces these methods to learn cross-view associations from extremely limited paired data, making it difficult to generalize to incomplete samples and thus hindering cross-view recovery. More critically, observation bias acts as a confounding factor, causing clustering predictions to become overly dependent on low-missing-rate views. Contrastive methods (Dong et al., 2025; Zhang et al., 2025) that learn cross-view mappings from abundant complete paired samples and treat all views equally consequently lead to clustering predictions that erroneously depend on extrinsic observation patterns rather than solely on the intrinsic data structure.

To overcome these limitations, we propose CIMLN, a novel Causal-Invariant Meta-Learning Network to alleviate observation bias for unbalanced incomplete multi-view clustering. Specifically, we introduce a context-aware meta-generation module that formulates missing view recovery as a meta-learning task. By encoding global statistical relationships through contextual information, this module enables the model to learn transferable generative meta-knowledge from scarce paired data and rapidly adapt to incomplete samples. Furthermore, the causal-invariant structure learning module in CIMLN identifies observation masks as confounding factors from a causal inference perspective and constructs counterfactual scenarios by randomly masking available views. This enforces consistency in clustering assignments between factual and counterfactual observation patterns, thereby eliminating the erroneous dependence of clustering on observation patterns. The meta-generation module provides high-quality missing view representations for counterfactual scenarios, while the causal-invariant module in turn imposes strong semantic constraints on the former. Through this deep synergy, CIMLN effectively alleviates observation bias in unbalanced incomplete multi-view scenarios. The main contributions are summarized as follows,

- We identify the observation bias problem under unbalanced incomplete multi-view scenarios along with its two derived challenges, namely difficulty in cross-view recovery and spurious dependence on observation patterns. We propose a causal-invariant meta-learning network to effectively mitigate observation bias.

- We innovatively integrate meta-learning and causal inference paradigms within CIMLN. Meta-learning overcomes information imbalance to enable rapid adaptation for missing views, while causal inference eliminates the erroneous dependence of clustering predictions on observation patterns.

- Extensive experiments on eight benchmarks demonstrate that CIMLN significantly outperforms existing methods under unbalanced missing scenarios.

## 2. Related Work

### 2.1. Deep Incomplete Multi-view Clustering

Deep neural networks improve clustering by learning nonlinear representations and modeling cross-view associations(Li et al., 2025a;b; Zhou et al., 2025; Wang et al., 2025a), which has motivated extensive research on deep incomplete multi-view clustering. A dominant paradigm is missing-view recovery(Chao et al., 2024; Lu et al., 2024; Feng et al., 2025), aiming to reconstruct missing data or latent representations to mitigate performance degradation. Existing methods mainly fall into three categories: (1) **GAN-based** approaches that synthesize missing views via adversarial learning, e.g., CPM-Nets(Zhang et al., 2022) and GP-MVC(Wang et al., 2021); (2) **Graph-based** approaches that transfer structural information to infer missing views, such as CRTC(Wang et al., 2023), SPCC(Dong et al., 2025), and ICMVC(Chao et al., 2024); and (3) **Entropy-based** approaches that minimize conditional entropy through dual-view prediction, including COMPLETER(Lin et al., 2021) and DCP(Lin et al., 2022). However, most of these methods implicitly assume relatively balanced missingness (or sufficient complete samples), and thus may be less effective under unbalanced missing patterns, where cross-view recovery generalizes poorly and clustering becomes biased toward low-missing-rate views.

### 2.2. Causal Representation Learning

Causal representation learning (Deng et al., 2025) seeks shift-robust features by enforcing causal invariance and using counterfactual reasoning to reduce confounder-induced spurious correlations. In multi-view learning (Fang et al., 2023), it views each modality as an observation of latent factors to separate shared semantics from view-specific noise. Recent attempts include identifiability under partial observability (Yao et al., 2023), mitigating noisy/dominant-view dependency (CausalMVC) (Bao et al., 2025), and intervention-based invariant clustering (CauMVC) (Yang

et al., 2025). However, they generally do not explicitly model observation bias under unbalanced missingness, where missingness patterns hinder cross-view transfer and spuriously bias clustering toward low-missing-rate views. We therefore treat missingness patterns as environmental variations and enforce observation-pattern-invariant clustering via counterfactual construction and invariant constraints.

## 3. Methodology

### 3.1. Motivation and Problem Formulation

In real-world scenarios, multi-view incompleteness often exhibits significant imbalance, causing observation bias and posing two core challenges for clustering: i) Existing methods struggle to learn reliable cross-view associations from imbalanced view information to recover missing instances; ii) Clustering predictions tend to erroneously depend on low-missing-rate views. Therefore, we propose a Causal-Invariant Meta-Learning Network to alleviate observation bias. The core modules of CIMLN are the View-specific Feature Extraction Module (VFS), the Context-aware Meta-generation Module (CM), and the Causal-invariant Structure Learning Module (CSL), as illustrated in Figure 2. The CM module learns transferable generative meta-knowledge from scarce paired data to enable high-quality recovery of missing views. The CSL module constructs counterfactual scenarios to eliminate the erroneous dependence of clustering on observation patterns.

We denote multi-view dataset as $\mathcal{D} = \{\mathbf{X}^1, \mathbf{X}^2, \cdots, \mathbf{X}^V\}$ with $N$ samples and $V$ views. For the $v$-th view, the data matrix is $\mathbf{X}^v = [\mathbf{x}_1^v, \mathbf{x}_2^v, \cdots, \mathbf{x}_N^v]^\top \in \mathbb{R}^{N \times d^v}$. The observation mask matrix $\mathbf{M} = [m_i^v] \in \{0,1\}^{N \times V}$ is defined where $m_i^v = 1$ indicates $\mathbf{x}_i^v$ is observed and $m_i^v = 0$ indicates missing. Our method aims to recover the missing instances and divide all samples into $K$ clusters.

### 3.2. View-specific Feature Extraction Module

For each view $v \in \{1, \ldots, V\}$, we define an encoder $\mathcal{E}_{\phi_v}^v : \mathbb{R}^{d^v} \to \mathbb{R}^{d_h}$ and a decoder $\mathcal{D}_{\psi_v}^v : \mathbb{R}^{d_h} \to \mathbb{R}^{d^v}$ that map raw features to a unified latent space of dimension $d_h$. The latent representation of sample $i$ in view $v$ is $\mathbf{h}_i^v = \mathcal{E}_{\phi_v}^v(\mathbf{x}_i^v)$, and the fused representation is $\mathbf{z}_i = [\mathbf{h}_i^1; \ldots; \mathbf{h}_i^V] \in \mathbb{R}^{V \cdot d_h}$. The reconstruction objective is defined as the normalized reconstruction error for observed samples,

$$\mathcal{L}_{rec} = \frac{1}{V} \sum_{v=1}^{V} \frac{\sum_{i=1}^{N} m_i^v \|\mathbf{x}_i^v - \mathcal{D}^v (\mathcal{E}^v(\mathbf{x}_i^v))\|_2^2}{\sum_{i=1}^{N} m_i^v}, \quad (1)$$

where $m_i^v \in \{0,1\}$ denotes the observation mask, with loss computed only for observed views. This formulation enables extraction of view-specific features that preserve semantic structure.

### 3.3. Context-aware Meta-generation Module

Inspired by meta-learning, we formulate missing view generation as a rapid adaptation task to generate target missing view representations from observed views. The complete sample set, comprising samples with all views observed, is defined as follows,

$$\mathcal{S}_c = \left\{ i \in \{1, \ldots, N\} \,\middle|\, \prod_{v=1}^{V} m_i^v = 1 \right\}. \quad (2)$$

$\mathcal{S}_c$ serves as the support set for meta-learning, providing complete cross-view mapping information. For sample $j \in \mathcal{S}_c$, the aggregated representation is defined as follows,

$$\bar{\mathbf{h}}_j = \frac{1}{V} \sum_{v=1}^{V} \mathbf{h}_j^v = \frac{1}{V} \sum_{v=1}^{V} \mathcal{E}_{\phi_v}^v(\mathbf{x}_j^v). \quad (3)$$

The meta-generator $\mathcal{G}_\theta$ learns the conditional distribution $p_\theta(\mathbf{h}^v | \bar{\mathbf{h}}, \mathcal{C}_v)$ to infer missing view representations. Context information $\mathcal{C}_v$ is defined as the expected second-order interactions over support samples:

$$\mathcal{C}_v = \mathbb{E}_{j \sim \mathcal{S}_c} [\bar{\mathbf{h}}_j \odot \mathbf{h}_j^v] = \frac{1}{|\mathcal{S}_c|} \sum_{j \in \mathcal{S}_c} \bar{\mathbf{h}}_j \odot \mathbf{h}_j^v, \quad (4)$$

where $\odot$ denotes the Hadamard product. This context encodes cross-view associations, enabling information from low-missing-rate views to guide the generation of high-missing-rate views. The target view representation follows a conditional Gaussian distribution:

$$p_\theta(\mathbf{h}^v | \bar{\mathbf{h}}, \mathcal{C}_v) = \mathcal{N}\left(\mathbf{h}^v; \boldsymbol{\mu}_\theta(\bar{\mathbf{h}}, \mathcal{C}_v), \mathrm{diag}(\boldsymbol{\sigma}_\theta^2(\bar{\mathbf{h}}, \mathcal{C}_v))\right) \quad (5)$$

where $\boldsymbol{\mu}_\theta$ and $\boldsymbol{\sigma}_\theta^2$ parameterize the mean and diagonal covariance, respectively. For high-missing-rate views, the model automatically increases variance to capture generation uncertainty.

Dimension normalization is introduced to eliminate the impact of latent space dimension on loss magnitude. For view $v$, with observed sample set $\mathcal{S}_v^{(o)} = \{i | m_i^v = 1\}$, the meta-learning loss is formulated as follows,

$$l_{\mathrm{meta}}^v = -\frac{1}{|\mathcal{S}_v^{(o)}|} \sum_{i \in \mathcal{S}_v^{(o)}} \frac{1}{d_h} \log p_\theta(\mathbf{h}_i^v | \bar{\mathbf{h}}_{i,-v}, \mathcal{C}_v), \quad (6)$$

where $\bar{\mathbf{h}}_{i,-v} = \frac{1}{|\mathcal{O}_i \setminus \{v\}|} \sum_{v' \in \mathcal{O}_i, v' \neq v} \mathbf{h}_i^{v'}$ and $\mathcal{O}_i = \{v | m_i^v = 1\}$ is the observed view set for sample $i$. Expanding the Gaussian log-likelihood with variance regularization as follows,

$$l_{meta}^v = \frac{1}{|\mathcal{S}_v^{(o)}|} \sum_{i \in \mathcal{S}_v^{(o)}} \left[ \frac{1}{d_h} \sum_{k=1}^{d_h} \frac{(\mathbf{h}_{i,k}^v - \mu_{\theta,k})^2}{\sigma_{\theta,k}^2} \right.$$
$$\left. + \lambda_{\mathrm{var}} \frac{1}{d_h} \sum_{k=1}^{d_h} \log(\sigma_{\theta,k}^2 + \epsilon) \right], \quad (7)$$

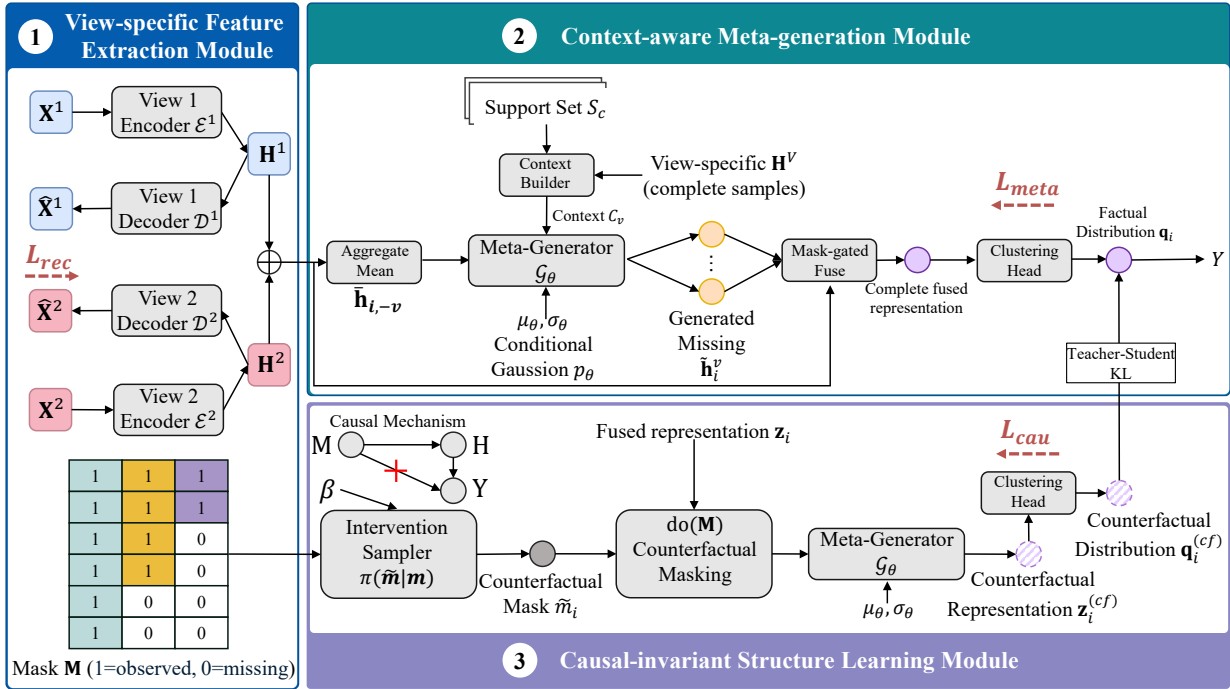

*Figure 2.* The framework of CIMLN. As shown, the core modules of CIMLN are the View-specific Feature Extraction Module (VFS), the Context-aware Meta-generation Module (CM), and the Causal-invariant Structure Learning Module (CSL). The VFS module projects the features of each view into a common space. CM module acquires transferable generative meta-knowledge from limited paired samples, which supports the high-fidelity recovery of missing views. Meanwhile, the CSL module builds counterfactual situations to remove the erroneous dependence of clustering on observation patterns.

where $\lambda_{\mathrm{var}}$ is the regularization coefficient and $\epsilon$ is the numerical stability constant. The final meta-learning loss is

$$\mathcal{L}_{meta} = \mathbb{E}_{v \sim \mathcal{U}(\mathcal{V}_{\mathrm{valid}})}\left[l^v_{meta}\right] = \frac{1}{|\mathcal{V}_{\mathrm{valid}}|}\sum_{v \in \mathcal{V}_{\mathrm{valid}}} l^v_{meta}, \quad (8)$$

where $\mathcal{V}_{\mathrm{valid}} = \{v \mid |\mathcal{S}^{(o)}_v| > 0\}$ is the valid view set.

For incomplete sample $i$ with missing view set $\mathcal{M}_i = \{v \mid m^v_i = 0\} \neq \emptyset$, the meta-generator generates missing view representations. For $v \in \mathcal{M}_i$, the generation process is

$$\tilde{\mathbf{h}}^v_i = \mathcal{G}_\theta(\bar{\mathbf{h}}_{i,-v}, \mathcal{C}_v) = \boldsymbol{\mu}_\theta(\bar{\mathbf{h}}_{i,-v}, \mathcal{C}_v). \quad (9)$$

During inference, the mean $\boldsymbol{\mu}_\theta$ is used as the generated representation for deterministic generation.

### 3.4. Causal-invariant Structure Learning Module

Under unbalanced missing scenarios, models tend to bias toward low-missing-rate views, causing clustering structures to spuriously depend on observation patterns. We introduce a structural causal model for analysis. Let $Y$ denote the true clustering labels, $\mathbf{H} = \{h^v\}^V_{v=1}$ the latent representations of all views, and $\mathbf{M} = \{m^v\}^V_{v=1}$ the observation masks. The observation mask $\mathbf{M}$ affects both $\mathbf{H}$ and clustering

prediction $\hat{Y}$, forming a confounding factor. The causal graph is $\mathbf{M} \to \mathbf{H} \to \hat{Y}$ with $\mathbf{M} \to \hat{Y}$.

To eliminate spurious dependence, intervention is performed to learn a causal-invariant clustering structure,

$$p(Y|\mathbf{H}, \mathrm{do}(\mathbf{M})) = p(Y|\mathbf{H}), \quad (10)$$

where $\mathrm{do}(\mathbf{M})$ denotes intervention on the observation pattern, breaking the spurious causal path between $\mathbf{M}$ and $Y$.

Counterfactual samples are constructed via random masking. For sample $i$ with factual observation mask $\mathbf{m}_i = [m^1_i, \cdots, m^V_i]^\top$, the intervention distribution is defined as,

$$\pi(\tilde{\mathbf{m}}_i|\mathbf{m}_i) = \prod^V_{v=1} \pi(\tilde{m}^v_i|m^v_i),$$

$$\pi(\tilde{m}^v_i|m^v_i) = \begin{cases} \mathrm{Bernoulli}(1-\beta), & \text{if } m^v_i = 1 \\ \delta_0(\tilde{m}^v_i), & \text{if } m^v_i = 0 \end{cases} \quad (11)$$

where $\beta \in (0,1)$ is the intervention strength and $\delta_0(\cdot)$ is the Dirac distribution. Observed views are masked with probability $\beta$, while missing views remain missing.

The fused representation for clustering assignment is con-

structed as

$$\mathbf{z}_i = \bigoplus_{v=1}^{V} \left( m_i^v \cdot \mathbf{h}_i^v + (1 - m_i^v) \cdot \tilde{\mathbf{h}}_i^v \right) \in \mathbb{R}^{V \cdot d_h} \quad (12)$$

where $\bigoplus$ denotes concatenation, with $\mathbf{h}_i^v$ for observed views and $\tilde{\mathbf{h}}_i^v$ for missing views. Given cluster center matrix $\mathbf{W} = [\mathbf{w}_1, \cdots, \mathbf{w}_K] \in \mathbb{R}^{V \cdot d_h \times K}$, the clustering assignment probability is:

$$\begin{aligned} q_{ik} &= p(y_i = k | \mathbf{z}_i, \mathbf{W}) \\ &= \frac{\exp\left(-\tau^{-1} \|\mathbf{z}_i - \mathbf{w}_k\|_2^2\right)}{\sum_{k'=1}^{K} \exp\left(-\tau^{-1} \|\mathbf{z}_i - \mathbf{w}_{k'}\|_2^2\right)}, \end{aligned} \quad (13)$$

where $\tau > 0$ is the temperature parameter. Denote the factual clustering assignment distribution as $\mathbf{q}_i = [q_{i1}, \cdots, q_{iK}]^\top$.

Given counterfactual mask $\tilde{\mathbf{m}}_i$, the counterfactual input is constructed as:

$$\mathbf{h}_i^{v,(cf)} = \mathbf{h}_i^v \cdot \mathbb{I}(\tilde{m}_i^v = 1), \quad (14)$$

where $\mathbb{I}(\cdot)$ is the indicator function. If a view is masked in the counterfactual scenario, its input is forced to zero. Using the meta-generator, the counterfactual fused representation $\mathbf{z}_i^{(cf)}$ is generated and the counterfactual clustering assignment is computed as:

$$\begin{aligned} q_{ik}^{(cf)} &= p(y_i = k | \mathbf{z}_i^{(cf)}, \mathbf{W}) \\ &= \frac{\exp\left(-\tau^{-1} \|\mathbf{z}_i^{(cf)} - \mathbf{w}_k\|_2^2\right)}{\sum_{k'=1}^{K} \exp\left(-\tau^{-1} \|\mathbf{z}_i^{(cf)} - \mathbf{w}_{k'}\|_2^2\right)}. \end{aligned} \quad (15)$$

Denote the counterfactual clustering assignment distribution as $\mathbf{q}_i^{(cf)} = [q_{i1}^{(cf)}, \cdots, q_{iK}^{(cf)}]^\top$.

KL divergence is used to minimize the difference between factual and counterfactual distributions. Treating the factual distribution $\mathbf{q}i$ as the teacher signal and the counterfactual distribution $\mathbf{q}_i^{(cf)}$ as the student signal, the causal loss is

$$\begin{aligned} \mathcal{L}_{cau} &= \mathbb{E}_{i \sim \mathcal{D}} \mathbb{E}_{\tilde{\mathbf{m}}_i \sim \pi(\cdot | \mathbf{m}_i)} \left[ D_{\mathrm{KL}}\left( \mathbf{q}_i \| \mathbf{q}_i^{(cf)} \right) \right] \\ &= \frac{1}{N} \sum_{i=1}^{N} \mathbb{E}_{\tilde{\mathbf{m}}_i \sim \pi(\cdot | \mathbf{m}_i)} \left[ \sum_{k=1}^{K} q_{ik} \log \frac{q_{ik}}{q_{ik}^{(cf)}} \right]. \end{aligned} \quad (16)$$

Minimizing this causal loss makes the generation mechanism causally robust, eliminating the spurious causal effect of observation patterns on clustering.

### 3.5. Objective Function and Optimization

In general, the objective loss function of CIMLN is formulated as follows,

$$\mathcal{L} = \mathcal{L}_{rec} + \lambda_1 \mathcal{L}_{meta} + \lambda_2 \mathcal{L}_{cau}, \quad (17)$$

**Algorithm 1** The optimization of CIMLN

**Input:** Dataset $\{\mathbf{X}^v\}_{v=1}^V$ of size $N$; number of clusters $K$; networks $\{\mathcal{E}_{\phi_v}^v, \mathcal{D}_{\psi_v}^v\}_{v=1}^V$ and $\mathcal{G}_\theta$; max epoch of two stage $E_1, E_2$; trade-off parameters $\lambda_1$ and $\lambda_2$.
**Output:** The predicted clustering results $Y$.
1: **for** $e = 1$ to $E_1$ **do**
2:      Update $\{\mathcal{E}_{\phi_v}^v, \mathcal{D}_{\psi_v}^v\}_{v=1}^V$ with Eq. (1)
3: **end for**
4: **for** $e = 1$ to $E_2$ **do**
5:      Compute aggregated representations for support samples via Eq. (3)
6:      Obtain context information for each view by Eq. (4)
7:      Predicts mean and variance by Eq. (5)
8:      Generate missing view representations via Eq. (9)
9:      Construct factual fused representation via Eq. (12)
10:      Compute factual clustering assignment via Eq. (13)
11:      Sample counterfactual mask via Eq. (11)
12:      Construct counterfactual input via Eq. (14)
13:      Compute counterfactual assignment via Eq. (15)
14:      Train the entire network with Eq. (17)
15: **end for**

where $\lambda_1$ and $\lambda_2$ are balanced coefficients used to adjust the influence of loss term. The three terms form a synergistic mechanism. $\mathcal{L}_{rec}$ prevents representation degradation, $\mathcal{L}_{meta}$ enables the meta-generator to learn effective cross-view mappings for recovering high-missing-rate views, and $\mathcal{L}_{cau}$ eliminates observation pattern bias via adversarial counterfactual scenarios, improving generalization and clustering consistency.

The optimization process of our CIMLN is listed in Algorithm 1. After optimization, the meta-generator generates missing view representations, which are concatenated with complete view representations, followed by k-means to obtain the final clustering result $Y$.

### 3.6. Theoretical Analysis

This section theoretically analyzes the applicability of CIMLN under unbalanced incomplete multi-view scenarios, providing rigorous mathematical guarantees.

**Theorem 3.1.** *Let the meta-generator $\mathcal{G}_\theta$ belong to hypothesis space $\mathcal{G}$ with empirical Rademacher complexity $\hat{\mathfrak{R}}_{|\mathcal{S}_c|}(\mathcal{G})$. For any view $v$ and any $\delta \in (0, 1)$, with probability at least $1 - \delta$, the expected generation error of the meta-generator on all samples satisfies:*

$$\begin{aligned} & \mathbb{E}_{(\bar{\mathbf{h}}, \mathbf{h}^v) \sim p_\mathcal{D}} \left[ \|\mathcal{G}_\theta(\bar{\mathbf{h}}, \mathcal{C}_v) - \mathbf{h}^v\|_2^2 \right] \\ & \leq \hat{l}_{meta}^v + 4 D_\mathcal{H}^2 \hat{\mathfrak{R}}_{|\mathcal{S}_c|}(\mathcal{G}) \\ & + D_\mathcal{H}^2 \sqrt{\frac{2 \log(2/\delta)}{|\mathcal{S}_c|}} + 4 D_\mathcal{H}^2 \cdot d_{TV}^{(v)}. \end{aligned} \quad (18)$$

where $\hat{l}_{meta}^v = \frac{1}{|\mathcal{S}_c|} \sum_{j \in \mathcal{S}_c} \|\mathcal{G}_\theta(\bar{\mathbf{h}}_j, \mathcal{C}_v) - \mathbf{h}_j^v\|_2^2$ is the empirical meta-learning loss.

Next, we analyze the theoretical guarantee of causal invariance. We establish the connection between causal invariance loss and clustering structure invariance based on the following theorem.

**Theorem 3.2.** *Let the causal invariance loss* $\mathcal{L}_{cau} = \mathbb{E}_i \mathbb{E}_{\tilde{\mathbf{m}}_i \sim \pi(\cdot|\mathbf{m}_i)}[D_{KL}(\mathbf{q}_i \| \mathbf{q}_i^{(cf)})] \leq \epsilon_c$, *where* $\mathbf{q}_i$ *is the factual clustering assignment and* $\mathbf{q}_i^{(cf)}$ *is the counterfactual clustering assignment. If the intervention distribution* $\pi$ *satisfies that for any factual mask* $\mathbf{m}$ *and target mask* $\mathbf{m}'$, *there exists a finite-length intervention chain* $\mathbf{m} = \mathbf{m}^{(0)} \to \mathbf{m}^{(1)} \to \cdots \to \mathbf{m}^{(L)} = \mathbf{m}'$ *where each step* $\mathbf{m}^{(l)} \to \mathbf{m}^{(l+1)}$ *has transition probability* $\pi(\mathbf{m}^{(l+1)}|\mathbf{m}^{(l)}) > 0$, *then the clustering assignment function is* $\epsilon$-*invariant:*

$$\epsilon \leq \sqrt{2\epsilon_c} \cdot \mathbb{E}_{\mathbf{m},\mathbf{m}' \sim p(\mathbf{M})}[L(\mathbf{m}, \mathbf{m}')], \qquad (19)$$

*where* $L(\mathbf{m}, \mathbf{m}')$ *is the shortest intervention chain length from* $\mathbf{m}$ *to* $\mathbf{m}'$.

Due to space constraints, detailed proofs of the above theorems are provided in the Appendix A.

# 4. Experiments

To validate the effectiveness of CIMLN, we conduct extensive experiments to answer the following questions: (Q1) Does CIMLN outperform current state-of-the-art methods in clustering performance on widely used datasets? (Q2) Do context-aware meta-generation and causal-invariant structure learning modules positively contribute to performance? (Q3) Can CIMLN recover high-quality representations of missing views? (Q4) How do the main hyperparameters affect CIMLN's performance?

## 4.1. Experimental Settings

### 4.1.1. DATASETS AND IMPLEMENTATION DETAILS

To thoroughly evaluate the effectiveness of the proposed CIMLN, we conduct experiments on eight widely used multi-view benchmark datasets, including CUB, Caltech, HandWritten, CiteSeer, Scene-15, Reuters, YouTubeFace20, and FashionMNIST. The detailed statistics of these datasets (e.g., the number of samples, clusters, views, and feature dimensionalities) are summarized in Table 2. In our experiments, the average missing rate (AMR) is set to 0.5. More details can be found in the Appendix B.

In experiments, we evaluate clustering performance using the three most common metrics for deep multi-view clustering, namely ACC, NMI, and ARI. Our model is implemented based on PyTorch 2.1.0 and trained on a desktop

computer configured with NVIDIA GeForce RTX 3090 and 64G RAM.

### 4.1.2. COMPARED METHODS

To demonstrate the performance of our proposed CIMLN, we compare it with twelve representative methods for incomplete multi-view clustering, including GP-MVC (Wang et al., 2021), DCP (Lin et al., 2022), DSIMVC (Tang & Liu, 2022), SURE (Yang et al., 2022), APADC (Xu et al., 2023), SMILE (Zeng et al., 2023), DIVIDE (Lu et al., 2024), DVIMC (Xu et al., 2024), PMIMC (Yuan et al., 2025), CPMN (Wang et al., 2025b), ESFMC (Chen et al., 2025b), and FIMCFG (Chao et al., 2025). For a fair comparison, all baselines are adapted to our experimental setting (i.e., the unbalanced missingness protocol) by following their original implementations and applying consistent preprocessing and missing-pattern generation.

## 4.2. Comparative Results Analysis (Q1)

We report the clustering results on eight multi-view benchmark datasets in Table 1, evaluated by three classic metrics. We compare our proposed CIMLN with twelve representative deep incomplete multi-view clustering methods, covering both recovery-based and recovery-free paradigms. From the results, we have the following observations:

- **Overall superiority and consistent improvements.** CIMLN achieves the best performance on all datasets in terms of ACC, demonstrating strong robustness under unbalanced missingness. Specifically, CIMLN improves the second-best method by 0.21%–4.62% points in ACC across datasets . Similar trends are observed for ARI, where CIMLN yields notable margins on challenging datasets, such as 4.71 on CiteSeer, 4.30 on YouTubeFace20, and 6.39 on FashionMNIST, indicating more reliable cluster assignments.

- **More pronounced gains on bias-sensitive datasets.** The advantages are particularly evident on datasets that are more vulnerable to observation-pattern bias. On CiteSeer, CIMLN achieves 56.55/31.93/27.95 (ACC/NMI/ARI), surpassing the strongest competitor by 4.62/4.01/4.71. This suggests that constructing counterfactual observations and enforcing causal-invariant consistency effectively mitigates spurious reliance on low-missing-rate views, thereby preserving clustering structure across varying observation patterns. On the large-scale YouTubeFace20, CIMLN further attains 76.74 ACC and 72.10 ARI, outperforming the best baseline by 2.32 and 4.30, respectively, which also reflects good scalability. We note that on Reuters, CIMLN remains the best in ACC/ARI while being slightly below the best method in NMI (by 0.21), im-

*Table 1.* : Clustering performance across eight multi-view benchmark datasets. The most outstanding results are denoted in **bold**, while the second-best values are underlined. The Average Missing Rate $AMR = 0.5$.

| Methods | Datasets | | | | | | | |
| --- | --- | --- | --- | --- | --- | --- | --- | --- |
| | CUB | Caltech | HandWritten | CiteSeer | Scene-15 | Reuters | YouTubeFace20 | FashionMNIST |
| | ACC(%) | | | | | | | |
| GP-MVC | 40.67 | 59.64 | 69.55 | 23.28 | 29.59 | 27.48 | 45.35 | 31.51 |
| DCP | 46.66 | 64.42 | 74.20 | 39.89 | 38.53 | 40.19 | 57.62 | 37.44 |
| DSIMVC | 45.67 | 63.61 | 71.29 | 33.76 | 25.89 | 39.19 | 54.95 | 35.53 |
| SURE | 53.80 | 77.22 | 72.40 | 32.61 | 39.14 | 40.44 | 52.96 | 40.82 |
| APADC | 39.50 | 67.82 | 68.34 | 37.44 | 33.10 | 36.80 | 51.13 | 38.66 |
| SMILE | 52.34 | 73.36 | 71.05 | 41.73 | 23.14 | 42.39 | 67.91 | 45.31 |
| DIVIDE | 58.26 | 81.57 | 75.45 | 43.36 | 37.73 | 40.90 | 68.92 | 49.18 |
| DVIMC | 47.83 | 83.00 | 80.57 | 35.08 | 33.29 | 36.08 | 59.37 | 50.78 |
| PMIMC | 53.27 | 83.64 | 86.80 | 44.19 | 40.58 | 38.26 | 72.52 | 55.80 |
| CPMN | 61.00 | 75.64 | 78.60 | 48.47 | 38.21 | 40.04 | 70.43 | 53.83 |
| ESFMC | 60.67 | 77.89 | 87.40 | 49.38 | 36.34 | 45.96 | 67.97 | 45.27 |
| FIMCFG | 62.17 | 82.29 | 86.65 | 51.93 | 40.00 | 48.08 | 74.42 | 51.89 |
| **CIMLN** | **62.83** | **83.93** | **90.30** | **56.55** | **41.29** | **48.29** | **76.74** | **56.33** |
| | NMI(%) | | | | | | | |
| GP-MVC | 47.04 | 46.89 | 61.80 | 13.96 | 30.89 | 12.39 | 58.39 | 26.39 |
| DCP | 54.89 | 51.79 | 72.34 | 20.64 | 40.56 | 21.64 | 60.50 | 36.42 |
| DSIMVC | 41.26 | 55.21 | 69.63 | 11.59 | 25.74 | 16.51 | 62.20 | 30.68 |
| SURE | 46.67 | 65.99 | 73.62 | 15.18 | 41.60 | 22.40 | 62.12 | 37.88 |
| APADC | 36.33 | 63.92 | 66.67 | 23.96 | 32.35 | 20.14 | 47.44 | 34.14 |
| SMILE | 48.09 | 64.38 | 65.30 | 25.34 | 22.72 | 23.78 | 73.86 | 54.67 |
| DIVIDE | 50.84 | 71.45 | 61.56 | 25.54 | 41.37 | 21.62 | 75.34 | 53.40 |
| DVIMC | 50.66 | 70.98 | 78.47 | 15.17 | 34.14 | 22.32 | 67.44 | 60.51 |
| PMIMC | 52.71 | 72.36 | 79.47 | 24.69 | 41.71 | 15.32 | 79.83 | 59.08 |
| CPMN | 59.57 | 69.12 | 63.91 | 21.67 | 39.55 | 18.28 | 80.59 | 56.58 |
| ESFMC | 58.28 | 67.89 | 80.40 | 27.92 | 36.11 | 21.29 | 77.64 | 53.06 |
| FIMCFG | 60.17 | 70.98 | 80.82 | 27.19 | 39.84 | **24.09** | 81.46 | 57.72 |
| **CIMLN** | **63.96** | **73.39** | **82.13** | **31.93** | **42.54** | 23.88 | **81.70** | **61.44** |
| | ARI(%) | | | | | | | |
| GP-MVC | 26.18 | 35.07 | 51.41 | 5.47 | 14.69 | 8.65 | 31.85 | 19.47 |
| DCP | 26.53 | 34.68 | 59.83 | 9.74 | 21.72 | 14.42 | 47.48 | 24.82 |
| DSIMVC | 25.79 | 43.20 | 57.58 | 8.96 | 12.34 | 14.79 | 45.56 | 25.05 |
| SURE | 33.14 | 60.08 | 62.19 | 8.76 | 22.87 | 10.59 | 41.50 | 27.21 |
| APADC | 24.10 | 54.23 | 55.79 | 11.97 | 17.81 | 15.08 | 37.39 | 24.13 |
| SMILE | 34.23 | 57.79 | 55.07 | 15.76 | 10.13 | 15.43 | 60.68 | 34.10 |
| DIVIDE | 36.82 | 59.20 | 52.02 | 14.00 | 23.12 | 15.81 | 60.50 | 30.45 |
| DVIMC | 34.08 | 67.88 | 71.15 | 9.71 | 13.43 | 10.65 | 45.85 | 37.69 |
| PMIMC | 36.79 | 68.80 | 75.02 | 13.47 | 23.36 | 14.90 | 69.48 | 37.16 |
| CPMN | 41.13 | 62.83 | 59.40 | 18.75 | 22.47 | 17.07 | 65.69 | 30.71 |
| ESFMC | 43.13 | 61.15 | 78.60 | 19.61 | 21.85 | 16.82 | 62.51 | 28.31 |
| FIMCFG | 45.38 | 66.89 | 75.02 | 23.24 | 20.45 | 17.66 | 67.80 | 35.67 |
| **CIMLN** | **46.44** | **69.11** | **79.97** | **27.95** | **24.15** | **18.14** | **72.10** | **44.08** |

*Table 2.* Incomplete multi-view datasets in experiments.

| Dataset | Samples | Clusters | Views | Dimensionality |
| --- | --- | --- | --- | --- |
| CUB | 600 | 10 | 2 | 1024/300 |
| Caltech | 1400 | 7 | 5 | 1984/512/928/254/40 |
| HandWritten | 2000 | 10 | 6 | 216/76/64/6/240/47 |
| CiteSeer | 3312 | 6 | 4 | 3703/3312/3312/3312 |
| Scene-15 | 4485 | 15 | 3 | 20/59/40 |
| Reuters | 18758 | 6 | 5 | 10/10/10/10/10 |
| YouTubeFace20 | 63896 | 20 | 4 | 944/576/512/640 |
| FashionMNIST | 70000 | 10 | 4 | 944/576/512/640 |

plying that the gain can be relatively smaller when intrinsic representations are already strong.

### 4.3. Ablation study (Q2)

We conduct ablation studies on eight benchmark datasets and report the results of three clustering metrics in Figure 3. We compare four configurations: removing all modules (None), using only the Context-aware Meta-generation Module (Only CM), using only the Causal-invariant Structure Learning Module (Only CSL), and the full model with both modules (Both). Overall, Both achieves the best performance on almost all datasets and metrics, indicating that CM and CSL are complementary. Compared with None, Only CM yields consistent improvements, suggest-

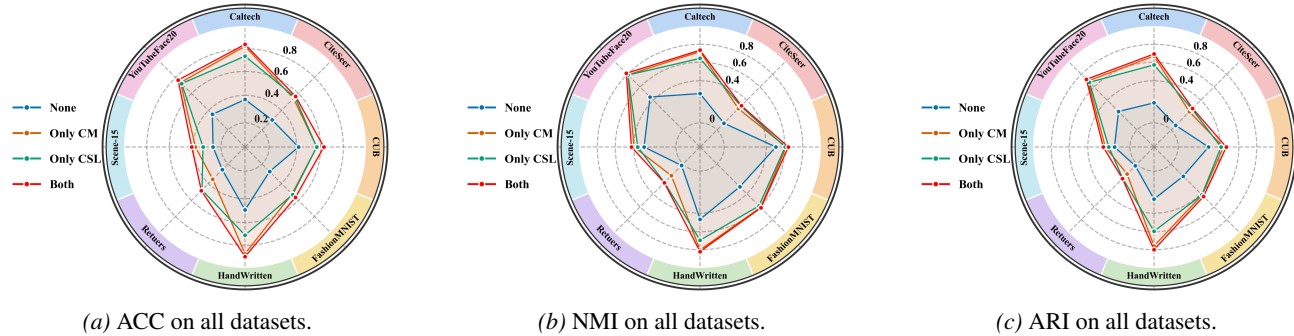

*(a)* ACC on all datasets.    *(b)* NMI on all datasets.    *(c)* ARI on all datasets.

*Figure 3.* The ablation results of CM and CSL modules on all datasets.

ing that context-aware meta-generation learns transferable cross-view relations from limited complete samples and alleviates the generalization difficulty of view recovery under unbalanced missingness. Only CSL also significantly outperforms None, implying that constructing counterfactual observations and enforcing causal-invariant consistency effectively suppresses observation bias induced by missingness patterns and reduces spurious reliance on low-missing-rate views. Furthermore, Both typically outperforms either single-module variant, demonstrating that unbalanced incomplete multi-view clustering requires jointly addressing generalizable view recovery and observation-pattern-invariant clustering structure.

### 4.4. Visualization of Recovery Effect (Q3)

To verify whether CIMLN can effectively reduce the discrepancy between recovered features and the underlying complete features under unbalanced missingness, we perform t-SNE visualization on the HandWritten dataset with an average missing rate of 0.5, as shown in Fig. 4. In the visualization, blue points denote samples with missing views (recovered features), while red points denote samples with complete views. As observed in Fig. 4a, the raw features exhibit an evident distribution gap between incomplete and complete samples, indicating that missingness induces notable representation shift. After recovery (Fig. 4b), the recovered samples become much better aligned with the complete ones within each cluster, and the gap is significantly reduced. These results suggest that CIMLN learns transferable cross-view associations and produces more consistent representations for incomplete samples, supporting its effectiveness in mitigating observation bias and improving clustering under unbalanced missingness.

### 4.5. Parameter Sensitivity Analysis (Q4)

We conduct experiments on HandWritten and CUB to study the sensitivity of CIMLN to the trade-off parameters $\lambda_1$ and $\lambda_2$. We vary $\lambda_1, \lambda_2 \in \{0.001, 0.01, 0.1, 1, 10\}$ and report ACC in Fig. 5. As shown in Fig. 5a, CIMLN achieves

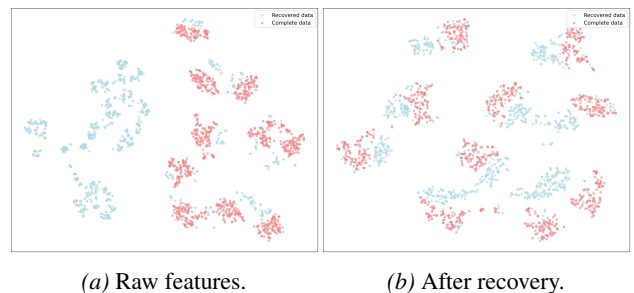

*(a)* Raw features.    *(b)* After recovery.

*Figure 4.* Visualization of the discrepancy between the recovered data and the ground truth on the HandWritten dataset, before and after training, under a average missing rate of 0.5.

consistently high performance on HandWritten across a broad range of settings, indicating low sensitivity. On CUB (Fig. 5b), the performance fluctuates more under extreme values, but remains stable and competitive for most combinations. Overall, CIMLN is robust to $\lambda_1$ and $\lambda_2$, requiring no delicate tuning in practice.

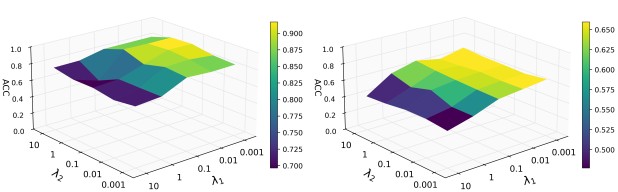

*(a)* ACC on HandWritten dataset.    *(b)* ACC on CUB dataset.

*Figure 5.* The ACC values yielded by the CIMLN method with different $\lambda_1$ and $\lambda_2$ combinations on CUB and HandWritten.

## 5. Conclusion

This paper proposes a causal-invariant meta-learning network to address the observation bias problem in unbalanced incomplete multi-view clustering. CIMLN innovatively integrates meta-learning and causal inference paradigms. The CM module learns transferable generative meta-knowledge

from scarce paired data for high-quality missing view recovery. The CSL module constructs counterfactual scenarios to eliminate the erroneous dependence of clustering on observation patterns. Through deep synergy between these two modules, CIMLN effectively alleviates observation bias. Experiments demonstrate superior performance of CIMLN in recovery fidelity and clustering tasks.

## Acknowledgements

This work was supported in part by the National Natural Science Foundation of China (No. 62325604, 62441618, 62506371, 62276271, 62406329, 62476280), the National Natural Science Foundation of China under Grant No. 62476281, the National Natural Science Foundation of China Joint Found under Grant No. U24A20323, the Major Program Project of Xiangjiang Laboratory (No. 24XJJ-CYJ01002), and the Brain Science and Brain-like Intelligence Technology-National Science and Technology Major Project (No. 2022ZD 0209100).

## Impact Statement

This paper presents work whose goal is to advance the field of Machine Learning. There are many potential societal consequences of our work, none which we feel must be specifically highlighted here.

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

# A. Theoretical Analysis and Proofs

This section provides a theoretical analysis of the applicability of CIMLN under the *unbalanced missingness* setting and establishes rigorous mathematical guarantees. We define the observation rate of view $v$ as $\rho^v = \frac{1}{N} \sum_{i=1}^N m_i^v$, where $m_i^v$ indicates whether the $i$-th sample is observed in view $v$. The degree of unbalanced missingness is quantified by the variance of the observation rates, $\sigma_\rho^2 = \frac{1}{V} \sum_{v=1}^V (\rho^v - \bar{\rho})^2$, $\bar{\rho} = \frac{1}{V} \sum_{v=1}^V \rho^v$, where $\bar{\rho}$ denotes the average observation rate across all views. A larger $\sigma_\rho^2$ indicates more severe unbalanced missingness.

## A.1. Generalization Error Bound of the Meta-Generator

We first analyze the generalization ability of the meta-generator on views with high missing rates.

**Assumption A.1** (Boundedness). The latent representation space $\mathcal{H} \subset \mathbb{R}^{d_h}$ is bounded, i.e., for all $\mathbf{h} \in \mathcal{H}$, $\|\mathbf{h}\|_2 \leq R$. We define the diameter of the latent space as

$$D_\mathcal{H} = \sup_{\mathbf{h}, \mathbf{h}' \in \mathcal{H}} \|\mathbf{h} - \mathbf{h}'\|_2 \leq 2R. \tag{20}$$

**Assumption A.2** (Lipschitz Continuity). The meta-generator $\mathcal{G}_\theta$ is $L_G$-Lipschitz continuous with respect to its input, i.e., for all $\mathbf{h}, \mathbf{h}' \in \mathcal{H}$,

$$\left\| \mathcal{G}_\theta(\mathbf{h}, \mathcal{C}_v) - \mathcal{G}_\theta(\mathbf{h}', \mathcal{C}_v) \right\|_2 \leq L_G \|\mathbf{h} - \mathbf{h}'\|_2. \tag{21}$$

**Definition A.3** (View-Conditional Distribution Discrepancy). For view $v$, we define the conditional distribution discrepancy between the complete-sample set $\mathcal{S}_c$ and the full dataset $\mathcal{D}$ in terms of the total variation distance as

$$d_{\text{TV}}^{(v)} = \sup_{\mathbf{h} \in \mathcal{H}} D_{\text{TV}} \left( p_{\mathcal{S}_c} \left( \mathbf{h}^v \mid \bar{\mathbf{h}} = \mathbf{h} \right), \ p_\mathcal{D} \left( \mathbf{h}^v \mid \bar{\mathbf{h}} = \mathbf{h} \right) \right). \tag{22}$$

**Theorem A.4** (Meta-Learning Generalization Error Bound). *Let the meta-generator $\mathcal{G}_\theta$ belong to a hypothesis class $\mathcal{G}$ with empirical Rademacher complexity $\hat{\mathfrak{R}}_{|\mathcal{S}_c|}(\mathcal{G})$. Under Assumptions (A.1, A.2), for any view $v$ and any $\delta \in (0, 1)$, with probability at least $1 - \delta$, the expected generation error of the meta-generator over the full dataset satisfies*

$$\mathbb{E}_{(\bar{\mathbf{h}}, \mathbf{h}^v) \sim p_\mathcal{D}} \left[ \left\| \mathcal{G}_\theta(\bar{\mathbf{h}}, \mathcal{C}_v) - \mathbf{h}^v \right\|_2^2 \right] \leq \hat{\mathcal{L}}_{\text{meta}}^{(v)} + 4 D_\mathcal{H}^2 \hat{\mathfrak{R}}_{|\mathcal{S}_c|}(\mathcal{G}) + D_\mathcal{H}^2 \sqrt{\frac{2 \log(2/\delta)}{|\mathcal{S}_c|}} + 4 D_\mathcal{H}^2 d_{\text{TV}}^{(v)}. \tag{23}$$

*Here,*

$$\hat{\mathcal{L}}_{\text{meta}}^{(v)} = \frac{1}{|\mathcal{S}_c|} \sum_{j \in \mathcal{S}_c} \left\| \mathcal{G}_\theta(\bar{\mathbf{h}}_j, \mathcal{C}_v) - \mathbf{h}_j^v \right\|_2^2, \tag{24}$$

*which denotes the empirical meta-learning loss.*

*Proof.* Define the loss function

$$\ell_\theta(\bar{\mathbf{h}}, \mathbf{h}^v) = \left\| \mathcal{G}_\theta(\bar{\mathbf{h}}, \mathcal{C}_v) - \mathbf{h}^v \right\|_2^2. \tag{25}$$

By Assumption A.1, the loss is bounded:

$$0 \leq \ell_\theta(\bar{\mathbf{h}}, \mathbf{h}^v) \leq D_\mathcal{H}^2. \tag{26}$$

**Step 1: Generalization bound under the complete-sample distribution.** Let the expected risk under the complete-sample distribution be $\mathcal{R}_{\mathcal{S}_c}(\theta) = \mathbb{E}_{p_{\mathcal{S}_c}}[\ell_\theta]$, and the empirical risk be $\hat{\mathcal{R}}_{\mathcal{S}_c}(\theta) = \hat{\mathcal{L}}_{\text{meta}}^{(v)}$. By the standard Rademacher-complexity generalization bound, for any $\delta \in (0, 1)$, with probability at least $1 - \delta$,

$$\mathcal{R}_{\mathcal{S}_c}(\theta) \leq \hat{\mathcal{R}}_{\mathcal{S}_c}(\theta) + 2 \hat{\mathfrak{R}}_{|\mathcal{S}_c|}(\ell \circ \mathcal{G}) + D_\mathcal{H}^2 \sqrt{\frac{2 \log(2/\delta)}{|\mathcal{S}_c|}}. \tag{27}$$

Since $\ell_\theta$ is $2D_\mathcal{H}$-Lipschitz with respect to the output of $\mathcal{G}_\theta$ (because $\left| \|\mathbf{a}\|_2^2 - \|\mathbf{b}\|_2^2 \right| \leq 2D_\mathcal{H} \|\mathbf{a} - \mathbf{b}\|_2$), the contraction property for Rademacher complexity yields

$$\hat{\mathfrak{R}}_{|\mathcal{S}_c|}(\ell \circ \mathcal{G}) \leq 2 D_\mathcal{H} \hat{\mathfrak{R}}_{|\mathcal{S}_c|}(\mathcal{G}). \tag{28}$$

Substituting the above inequality gives

$$\mathcal{R}_{\mathcal{S}_c}(\theta) \leq \hat{\mathcal{L}}_{\mathrm{meta}}^{(v)} + 4D_{\mathcal{H}}^2 \,\hat{\mathfrak{R}}_{|\mathcal{S}_c|}(\mathcal{G}) + D_{\mathcal{H}}^2 \sqrt{\frac{2\log(2/\delta)}{|\mathcal{S}_c|}}. \tag{29}$$

**Step 2: Effect of distribution shift.** Define the expected risk over the full dataset as $\mathcal{R}_{\mathcal{D}}(\theta) = \mathbb{E}_{p_{\mathcal{D}}}[\ell_\theta]$. The error induced by distribution shift can be bounded as

$$|\mathcal{R}_{\mathcal{D}}(\theta) - \mathcal{R}_{\mathcal{S}_c}(\theta)| = \left| \int_{\mathcal{H}}\!\!\int_{\mathcal{H}} \ell_\theta(\bar{\mathbf{h}}, \mathbf{h}^v) \left( p_{\mathcal{D}}(\bar{\mathbf{h}}, \mathbf{h}^v) - p_{\mathcal{S}_c}(\bar{\mathbf{h}}, \mathbf{h}^v) \right) d\bar{\mathbf{h}}\,d\mathbf{h}^v \right|. \tag{30}$$

Using the conditional factorization $p(\bar{\mathbf{h}}, \mathbf{h}^v) = p(\mathbf{h}^v \mid \bar{\mathbf{h}})\,p(\bar{\mathbf{h}})$, and assuming the marginal distributions are close, $p_{\mathcal{D}}(\bar{\mathbf{h}}) \approx p_{\mathcal{S}_c}(\bar{\mathbf{h}})$ (which holds when the complete-sample set is sufficiently large), we obtain

$$|\mathcal{R}_{\mathcal{D}}(\theta) - \mathcal{R}_{\mathcal{S}_c}(\theta)| \leq \int_{\mathcal{H}} p(\bar{\mathbf{h}}) \left| \int_{\mathcal{H}} \ell_\theta(\bar{\mathbf{h}}, \mathbf{h}^v) \left( p_{\mathcal{D}}(\mathbf{h}^v \mid \bar{\mathbf{h}}) - p_{\mathcal{S}_c}(\mathbf{h}^v \mid \bar{\mathbf{h}}) \right) d\mathbf{h}^v \right| d\bar{\mathbf{h}}. \tag{31}$$

By the definition of total variation distance and the boundedness of the loss, the above term is further bounded by

$$|\mathcal{R}_{\mathcal{D}}(\theta) - \mathcal{R}_{\mathcal{S}_c}(\theta)| \leq 4D_{\mathcal{H}}^2 \, d_{\mathrm{TV}}^{(v)}. \tag{32}$$

**Step 3: Combining the two parts.** Combining Step 1 and Step 2 yields

$$\mathcal{R}_{\mathcal{D}}(\theta) \leq \mathcal{R}_{\mathcal{S}_c}(\theta) + 4D_{\mathcal{H}}^2 \, d_{\mathrm{TV}}^{(v)}$$

$$\leq \hat{\mathcal{L}}_{\mathrm{meta}}^{(v)} + 4D_{\mathcal{H}}^2 \,\hat{\mathfrak{R}}_{|\mathcal{S}_c|}(\mathcal{G}) + D_{\mathcal{H}}^2 \sqrt{\frac{2\log(2/\delta)}{|\mathcal{S}_c|}} + 4D_{\mathcal{H}}^2 \, d_{\mathrm{TV}}^{(v)}. \tag{33}$$

This completes the proof. $\qquad\square$

*Remark* A.5. Theorem A.4 indicates that the generalization error of the meta-generator consists of four components: the empirical risk, the model complexity term, the sampling error, and the distribution-shift term. Even when the observation rate $\rho^v$ of view $v$ is low, effective generalization guarantees can still be obtained provided that the complete-sample set size $|\mathcal{S}_c|$ is sufficiently large and the distribution discrepancy $d_{\mathrm{TV}}^{(v)}$ is bounded. Moreover, the contextual information $\mathcal{C}_v$, by capturing global second-order statistics, helps reduce the conditional distribution discrepancy $d_{\mathrm{TV}}^{(v)}$.

### A.2. Theoretical Guarantee of Causal Invariance

We establish a theoretical connection between the causal-invariance loss and the invariance of the clustering structure.

**Definition A.6** (Observation-Pattern Invariance of Cluster Assignments). A clustering assignment function $\mathcal{C}$ is said to be $\epsilon$-invariant with respect to the observation pattern $\mathbf{M}$ if, for any sample $i$ and any feasible pair of observation patterns $(\mathbf{m}, \mathbf{m}')$,

$$D_{\mathrm{TV}}\left( \mathbf{q}_i^{(\mathbf{m})}, \mathbf{q}_i^{(\mathbf{m}')} \right) \leq \epsilon, \tag{34}$$

where $\mathbf{q}_i^{(\mathbf{m})} = \left[ q_{i1}^{(\mathbf{m})}, \ldots, q_{iK}^{(\mathbf{m})} \right]^\top$ denotes the cluster-assignment probability vector of sample $i$ under pattern $\mathbf{m}$, and

$$D_{\mathrm{TV}}(\mathbf{p}, \mathbf{q}) = \frac{1}{2} \sum_{k=1}^{K} |p_k - q_k| \tag{35}$$

is the total variation distance.

**Lemma A.7** (Pinsker's Inequality). *For any two probability distributions $\mathbf{p}$ and $\mathbf{q}$,*

$$D_{\mathrm{TV}}(\mathbf{p}, \mathbf{q}) \leq \sqrt{\frac{1}{2} D_{\mathrm{KL}}(\mathbf{p} \,\|\, \mathbf{q})}. \tag{36}$$

**Theorem A.8** (Causal-Invariance Loss Implies Clustering Invariance). *Suppose the causal-invariance loss satisfies*

$$\mathcal{L}_{\text{causal}} = \mathbb{E}_i \, \mathbb{E}_{\tilde{\mathbf{m}}_i \sim \pi(\cdot | \mathbf{m}_i)} \Big[ D_{\text{KL}}\big(\mathbf{q}_i \, \| \, \mathbf{q}_i^{(\text{cf})}\big) \Big] \leq \epsilon_c, \tag{37}$$

*where $\mathbf{q}_i$ denotes the factual cluster assignment and $\mathbf{q}_i^{(\text{cf})}$ denotes the counterfactual cluster assignment. Assume the intervention distribution $\pi$ satisfies the following connectivity condition: for any factual mask $\mathbf{m}$ and any target mask $\mathbf{m}'$, there exists a finite-length intervention chain*

$$\mathbf{m} = \mathbf{m}^{(0)} \to \mathbf{m}^{(1)} \to \cdots \to \mathbf{m}^{(L)} = \mathbf{m}', \tag{38}$$

*such that each transition has positive probability, i.e., $\pi\big(\mathbf{m}^{(l+1)} \mid \mathbf{m}^{(l)}\big) > 0$ for all $l$. Then the clustering assignment function is $\epsilon$-invariant, with*

$$\epsilon \leq \sqrt{2\epsilon_c} \, \mathbb{E}_{\mathbf{m}, \, \mathbf{m}' \sim p(\mathbf{M})} \big[ L(\mathbf{m}, \mathbf{m}') \big], \tag{39}$$

*where $L(\mathbf{m}, \mathbf{m}')$ denotes the length of the shortest intervention chain from $\mathbf{m}$ to $\mathbf{m}'$.*

*Proof.* **Step 1: A total-variation bound for one-step interventions.** For a one-step intervention $\mathbf{m} \to \tilde{\mathbf{m}}$, by the definition of the causal-invariance loss,

$$\mathbb{E}_{i, \, \tilde{\mathbf{m}} \sim \pi(\cdot | \mathbf{m})} \Big[ D_{\text{KL}}\big(\mathbf{q}_i^{(\mathbf{m})} \, \| \, \mathbf{q}_i^{(\tilde{\mathbf{m}})}\big) \Big] \leq \epsilon_c. \tag{40}$$

By Pinsker's inequality and Jensen's inequality (since $\sqrt{\cdot}$ is concave),

$$\mathbb{E}_{i, \, \tilde{\mathbf{m}}} \Big[ D_{\text{TV}}\big(\mathbf{q}_i^{(\mathbf{m})}, \mathbf{q}_i^{(\tilde{\mathbf{m}})}\big) \Big] \leq \mathbb{E}_{i, \, \tilde{\mathbf{m}}} \left[ \sqrt{\frac{1}{2} D_{\text{KL}}\big(\mathbf{q}_i^{(\mathbf{m})} \, \| \, \mathbf{q}_i^{(\tilde{\mathbf{m}})}\big)} \right]$$

$$\leq \sqrt{\frac{1}{2} \mathbb{E}_{i, \, \tilde{\mathbf{m}}} \Big[ D_{\text{KL}}\big(\mathbf{q}_i^{(\mathbf{m})} \, \| \, \mathbf{q}_i^{(\tilde{\mathbf{m}})}\big) \Big]} \leq \sqrt{\frac{\epsilon_c}{2}}. \tag{41}$$

A tighter bound can be obtained by applying Pinsker's inequality directly to the (expected) KL divergence, yielding

$$\mathbb{E}_{i, \, \tilde{\mathbf{m}}} \Big[ D_{\text{TV}}\big(\mathbf{q}_i^{(\mathbf{m})}, \mathbf{q}_i^{(\tilde{\mathbf{m}})}\big) \Big] \leq \sqrt{2\epsilon_c}. \tag{42}$$

**Step 2: Accumulation over multi-step intervention chains.** For an intervention chain from $\mathbf{m}$ to $\mathbf{m}'$, $\mathbf{m} = \mathbf{m}^{(0)} \to \mathbf{m}^{(1)} \to \cdots \to \mathbf{m}^{(L)} = \mathbf{m}'$, the triangle inequality of total variation distance gives

$$D_{\text{TV}}\big(\mathbf{q}_i^{(\mathbf{m})}, \mathbf{q}_i^{(\mathbf{m}')}\big) \leq \sum_{l=0}^{L-1} D_{\text{TV}}\big(\mathbf{q}_i^{(\mathbf{m}^{(l)})}, \mathbf{q}_i^{(\mathbf{m}^{(l+1)})}\big). \tag{43}$$

Taking expectations on both sides yields

$$\mathbb{E}_i \Big[ D_{\text{TV}}\big(\mathbf{q}_i^{(\mathbf{m})}, \mathbf{q}_i^{(\mathbf{m}')}\big) \Big] \leq \sum_{l=0}^{L-1} \mathbb{E}_i \Big[ D_{\text{TV}}\big(\mathbf{q}_i^{(\mathbf{m}^{(l)})}, \mathbf{q}_i^{(\mathbf{m}^{(l+1)})}\big) \Big] \leq L \sqrt{2\epsilon_c}. \tag{44}$$

**Step 3: Averaging over all mask pairs.** Averaging over all feasible mask pairs $(\mathbf{m}, \mathbf{m}')$ gives

$$\mathbb{E}_{\mathbf{m}, \, \mathbf{m}'} \mathbb{E}_i \Big[ D_{\text{TV}}\big(\mathbf{q}_i^{(\mathbf{m})}, \mathbf{q}_i^{(\mathbf{m}')}\big) \Big] \leq \mathbb{E}_{\mathbf{m}, \, \mathbf{m}'} \big[ L(\mathbf{m}, \mathbf{m}') \big] \sqrt{2\epsilon_c}. \tag{45}$$

By Definition 2, $\epsilon$-invariance requires the above bound to hold uniformly over $(\mathbf{m}, \mathbf{m}')$, hence

$$\epsilon \leq \sqrt{2\epsilon_c} \, \mathbb{E}_{\mathbf{m}, \, \mathbf{m}'} \big[ L(\mathbf{m}, \mathbf{m}') \big]. \tag{46}$$

This completes the proof. □

*Remark* A.9. Theorem 2 establishes a quantitative relationship between the causal-invariance loss $\epsilon_c$ and the clustering invariance level $\epsilon$. As $\epsilon_c \to 0$, we have $\epsilon \to 0$, implying that the cluster assignments become fully invariant to observation patterns. The shortest intervention-chain length $L(\mathbf{m}, \mathbf{m}')$ depends on the design of the intervention policy; under our random masking strategy, one has $L(\mathbf{m}, \mathbf{m}') \leq V$ (at most $V$ views need to be masked).

# B. Experimental Details

## B.1. Datasets

**CUB**[1] is a bird classification dataset using the first 10 categories, with two views: GoogLeNet(Szegedy et al., 2015) deep visual features and doc2vec(Le & Mikolov, 2014) textual representations.

**Caltech**(Xu et al., 2022) is an object image dataset with 1,400 samples from 7 categories, using five views extracted by different deep backbones: 1,984-dim features from VGG16, 512-dim from ResNet50, 928-dim from GoogLeNet, 254-dim from InceptionV3, and 40-dim from handcrafted Gabor features.

**HandWritten**[2] includes digits '0' to '9', with 200 samples per class, using six views: profile correlations, Fourier coefficients, Karhunen-Loeve coefficients, morphological, pixel features, and Zernike moments.

**CiteSeer**[3] is a graph dataset with 3,312 samples in 6 categories. We use cites, content, inbound, and outbound as four view features instead of the adjacency graph.

**Scene-15**(Zhang et al., 2017) is a scene recognition dataset with 4,485 images from 15 categories, using three views: 20-dim GIST descriptors, 59-dim PHOG features, and 40-dim LBP features.

**Reuters**[4] is a text dataset with 18,758 samples in 6 categories, using documents in German, English, French, Italian, and Spanish as views.

**YouTubeFace20**(Huang et al., 2023) contains 20 person categories with 38,654 samples, using features from video frames: LBP, HOG, GIST, and Gabor.

**FashionMNIST**[5] is a clothing image dataset with 70,000 samples from 10 categories. Following common multi-view settings, we construct four views using deep features extracted by different CNN backbones, yielding 944/576/512/640-dimensional representations, respectively.

## B.2. Compared Methods

We compare CIMLN with twelve representative incomplete multi-view clustering methods.

**GP-MVC** (Wang et al., 2021) adopts a CycleGAN-style framework to synthesize missing views via cycle-consistent cross-view generation.

**DCP** (Lin et al., 2022) learns consistent representations by jointly maximizing mutual information and minimizing a dual-view prediction loss based on cross-entropy.

**DSIMVC** (Tang & Liu, 2022) performs deep incomplete multi-view clustering by learning view-consistent latent representations and leveraging similarity/structure constraints to bridge missing views.

**SURE** (Yang et al., 2022) designs a noise-robust contrastive objective to alleviate the impact of false negatives in incomplete multi-view representation learning.

**APADC** (Xu et al., 2023) proposes an imputation-free alignment strategy that matches multi-view distributions in a shared latent space for clustering.

**SMILE** (Zeng et al., 2023) captures view-invariant semantic distributions without requiring paired samples, enabling clustering under incomplete correspondence.

**DIVIDE** (Lu et al., 2024) reduces false negatives by using high-order random walks to identify reliable in-cluster neighbors for contrastive learning.

**DVIMC** (Xu et al., 2024) avoids explicit recovery by optimizing per-view embeddings with an EM-like procedure to infer clustering assignments.

---

[1] http://www.vision.caltech.edu/visipedia/CUB-200.html

[2] https://archive.ics.uci.edu/ml/datasets/Multiple+Features

[3] https://linqs-data.soe.ucsc.edu/public/lbc/citeseer.tgz

[4] https://archive.ics.uci.edu/ml/datasets.html

[5] https://github.com/zalandoresearch/fashion-mnist

*Table 3.* AMR settings under different numbers of views. In our experiments, the average missing rate (AMR) is set to 0.5.

| view number | view1 | view2 | view3 | view4 | view5 | view6 |
|:---:|:---:|:---:|:---:|:---:|:---:|:---:|
| 2 | 1xAMR | 0xAMR | – | – | – | – |
| 3 | 1xAMR | 0.5xAMR | 0xAMR | – | – | – |
| 4 | 1xAMR | 0.75xAMR | 0.25xAMR | 0xAMR | – | – |
| 5 | 1xAMR | 0.75xAMR | 0.5xAMR | 0.25xAMR | 0xAMR | – |
| 6 | 1xAMR | 0.8xAMR | 0.6xAMR | 0.4xAMR | 0.2xAMR | 0xAMR |

**PMIMC** (Yuan et al., 2025) introduces prototype-level regularization to align incomplete views, encouraging samples to agree on shared cluster prototypes.

**CPMN** (Wang et al., 2025b) learns cross-view completion through a pretraining-and-matching scheme, where missing views are reconstructed to preserve clustering structure.

**ESFMC** (Chen et al., 2025b) enhances incomplete multi-view clustering by selectively fusing reliable semantic information across views and suppressing noisy/inconsistent cues.

**FIMCFG** (Chao et al., 2025) studies incomplete multi-view clustering in a federated setting and proposes a collaborative optimization strategy to handle view-missing data across clients.

### B.3. Data preprocessing

To simulate unbalanced incomplete multi-view data, we follow a controlled missing-rate protocol based on the average missing rate (AMR). Given an AMR value, we assign each view a view-specific missing rate as a fixed proportion of AMR, so that different views have distinct missingness levels while their overall average is governed by AMR. Table 3 summarizes the adopted missing-rate multipliers under different numbers of views. Specifically, for 3 views, the missing rates are set to $1 \times \text{AMR}$, $0.5 \times \text{AMR}$, and $0 \times \text{AMR}$; For 6 views, they are $1 \times \text{AMR}$, $0.8 \times \text{AMR}$, $0.6 \times \text{AMR}$, $0.4 \times \text{AMR}$, $0.2 \times \text{AMR}$, and $0 \times \text{AMR}$. In our experiments, we set $\text{AMR} = 0.5$, and the corresponding view-wise missing rates are obtained accordingly. This unbalanced design makes low-missing-rate views more dominant and induces observation bias, which matches our problem setting.

### B.4. Additional ablation results.

To provide exact numbers for reproducibility, Table 4 reports the numerical ACC/NMI/ARI results corresponding to "Ablation study (Q2)" under AMR= 0.5, using the same four variants (Without, Only $\mathcal{L}_{meta}$, Only $\mathcal{L}_{cau}$, and Both).

### B.5. Parameter Sensitivity Analysis.

**Parameter Sensitivity Analysis.** To further examine the robustness of CIMLN to the trade-off hyperparameters, we conduct a sensitivity study on six datasets (Caltech, CiteSeer, FashionMNIST, Reuters, Scene-15, and YouTubeFace20). Fig. 6 visualizes the clustering performance (ACC) under different $\lambda_1$–$\lambda_2$ combinations, where $\lambda_1$ and $\lambda_2$ control the relative strengths of the major objectives in CIMLN.

Overall, the ACC surfaces are smooth on most datasets and remain at a high level over a broad range of settings, indicating that CIMLN is not overly sensitive to $\lambda_1$ and $\lambda_2$. In particular, Caltech and YouTubeFace20 exhibit relatively stable performance across many combinations, suggesting that the proposed objective can be optimized reliably without delicate tuning. On CiteSeer, Reuters, and Scene-15, larger fluctuations mainly occur at extreme values of $\lambda_1$ or $\lambda_2$, while moderate settings consistently yield competitive results. This behavior is expected in unbalanced incomplete multi-view clustering, where overly emphasizing a single term may weaken either the generalizability of view recovery or the invariance to observation patterns. In practice, choosing $\lambda_1$ and $\lambda_2$ from a moderate range (e.g., $10^{-2}$ to 1) generally provides stable performance across datasets.

| Model | Datasets | | | | | | | |
|---|---|---|---|---|---|---|---|---|
| | Caltech | CiteSeer | CUB | FashionMNIST | HandWritten | Reuters | Scene-15 | YouTubeFace20 |
| **ACC** | | | | | | | | |
| Without | 0.3664 | 0.2953 | 0.4150 | 0.2696 | 0.4925 | 0.2511 | 0.2497 | 0.3564 |
| Only $\mathcal{L}_{meta}$ | 0.8229 | 0.5405 | 0.5667 | 0.5223 | 0.8770 | 0.3520 | 0.3837 | 0.7364 |
| Only $\mathcal{L}_{cau}$ | 0.7336 | 0.5571 | 0.5667 | 0.5298 | 0.7100 | 0.4753 | 0.3237 | 0.7232 |
| Both | **0.8393** | **0.5655** | **0.6283** | **0.5633** | **0.9030** | **0.4829** | **0.4129** | **0.7674** |
| **NMI** | | | | | | | | |
| Without | 0.2678 | 0.0777 | 0.5030 | 0.2972 | 0.4648 | 0.0142 | 0.2936 | 0.4479 |
| Only $\mathcal{L}_{meta}$ | 0.7181 | 0.2759 | 0.6155 | 0.6088 | 0.8016 | 0.1412 | 0.3981 | 0.7916 |
| Only $\mathcal{L}_{cau}$ | 0.6423 | 0.3126 | 0.6036 | 0.5844 | 0.6939 | 0.2365 | 0.3591 | 0.7899 |
| Both | **0.7339** | **0.3193** | **0.6396** | **0.6144** | **0.8213** | **0.2388** | **0.4254** | **0.8170** |
| **ARI** | | | | | | | | |
| Without | 0.1757 | 0.0505 | 0.2791 | 0.1482 | 0.2546 | 0.0172 | 0.1317 | 0.2356 |
| Only $\mathcal{L}_{meta}$ | 0.6687 | 0.2328 | 0.4347 | 0.4189 | 0.7580 | 0.1189 | 0.2209 | 0.6955 |
| Only $\mathcal{L}_{cau}$ | 0.5709 | 0.2706 | 0.4082 | 0.4160 | 0.5948 | 0.1772 | 0.1861 | 0.6741 |
| Both | **0.6911** | **0.2795** | **0.4644** | **0.4408** | **0.7997** | **0.1814** | **0.2415** | **0.7210** |

*Table 4.* Ablation study of different loss components on eight datasets. We report ACC/NMI/ARI, and the best results are highlighted in **bold**. The average missing rate AMR=0.5.

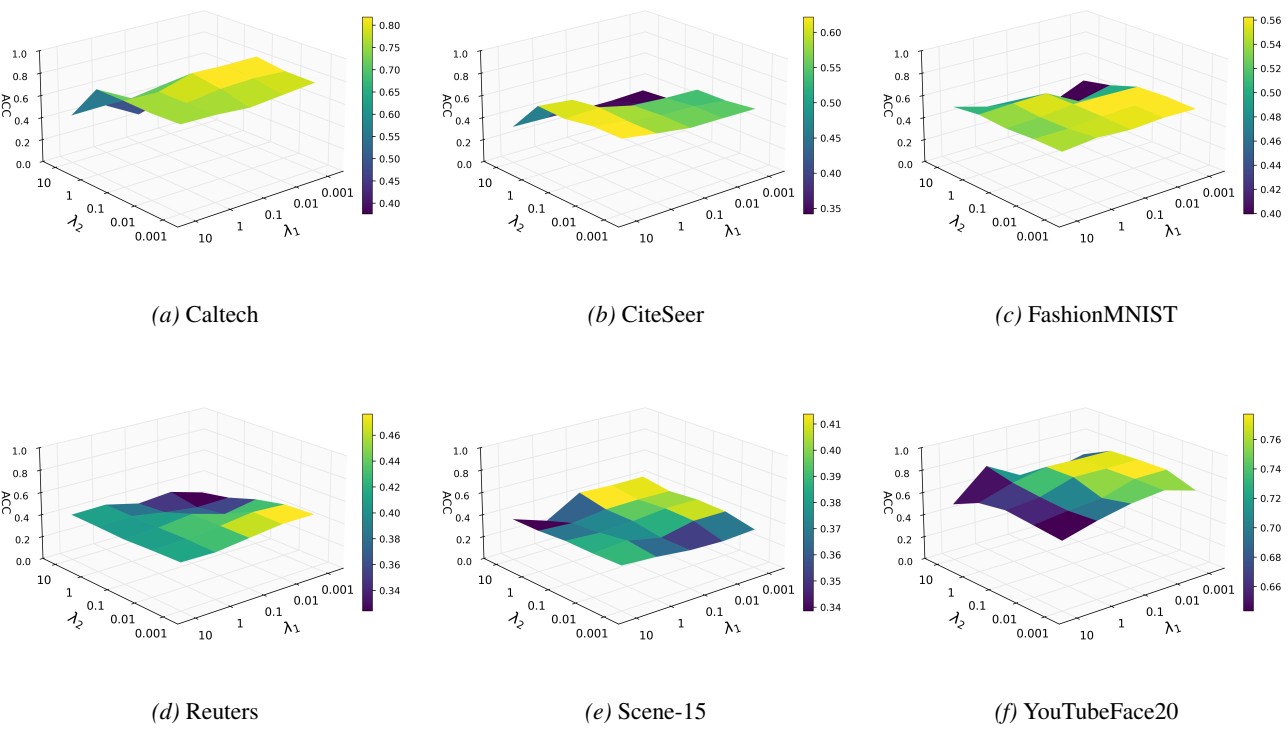

*(a)* Caltech      *(b)* CiteSeer      *(c)* FashionMNIST

*(d)* Reuters      *(e)* Scene-15      *(f)* YouTubeFace20

*Figure 6.* Parameter sensitivity analysis of CIMLN on six datasets (ACC under different $\lambda_1 - \lambda_2$ combinations).

