# OpenReview forum: "Alleviating Observation Bias via Causal-Invariant Meta-Learning for Unbalanced Incomplete Multi-view Clustering"
_ICML.cc/2026/Conference — ICML 2026 regular_

### Official Review · Reviewer_uZXW · 2026-03-04

**Soundness:** 2
**Presentation:** 2
**Significance:** 2
**Originality:** 2
**Overall Recommendation:** 2
**Confidence:** 5

**Summary:**

This paper addresses the problem of Unbalanced Incomplete Multi-View Clustering (UIMVC), where different views of a dataset experience significantly varying missingness rates. The authors propose CIMLN, a Causal-Invariant Meta-Learning Network. The architecture consists of three main components: a View-Specific Feature Extraction Module for baseline encoding, a Context-aware Meta-generation Module (CM) that formulates the missing view recovery as a meta-learning adaptation task trained on complete sample pairs, and a Causal-invariant Structure Learning Module (CSL) that constructs counterfactual missing scenarios by artificially masking available views. The stated goal is to prevent the clustering algorithm from spuriously depending on views with low missing rates. The model is evaluated on eight benchmark datasets, reporting improvements over several deep incomplete multi-view clustering methods.

**Compliance With Llm Reviewing Policy:**

Affirmed.

**Key Questions For Authors:**

-How can you mathematically justify the assumption $p_{\mathcal{D}}(\overline{h}) \approx p_{\mathcal{S}_{c}}(\overline{h})$ in your theoretical analysis when real-world unbalanced missingness almost guarantees a severe distribution shift between fully observed and partially observed cohorts?

-Detail the exact size of the support set $\mathcal{S}_c$ for each dataset under your experimental settings. At what minimum threshold of $|\mathcal{S}_c|$ does your CM module catastrophically fail to learn a valid conditional distribution?

-Provide a rigorous, mathematical distinction between your "causal-invariant counterfactual masking" and standard feature dropout with consistency regularization. Stripped of the causal terminology, what is the actual algorithmic difference?

-How exactly were the baseline methods modified to handle the UIMVC protocol? If they were not explicitly designed for this, is it not expected that they would fail, rendering your comparative baselines fundamentally weak?

**Limitations:**

The authors entirely fail to address the critical limitations of their method. The most glaring omission is the failure to discuss the model's absolute dependence on a sufficient quantity of fully complete multi-view samples ($\mathcal{S}_c$) to train the meta-generator. By ignoring this, the authors attempt to sidestep the exact problem they claim to solve: data sparsity in unbalanced scenarios. Failure to explicitly state the breakdown point of their algorithm relative to the size of $\mathcal{S}_c$ is a major oversight that invalidates the practical utility of the method.

**Strengths And Weaknesses:**

It is acknowledged that studying multi-view clustering under imbalanced missingness patterns is a relevant and practical application area. The initial motivation to decouple the clustering mechanism from the extrinsic observation pattern is conceptually interesting.However, as we dive deeper into the methodology, the facade of a novel contribution crumbles rapidly. The structural and theoretical claims made here are profoundly weak.First, the core premise of the Context-aware Meta-generation Module is structurally flawed. The authors claim to solve the problem of scarce paired data under unbalanced missingness. However, the CM module is trained exclusively on the support set $\mathcal{S}_c$ (the complete samples). If the missingness is truly severe and unbalanced as claimed in the introduction (e.g., medical imaging at <20% availability), the size of $\mathcal{S}_c$ will be trivially small. It is actually observed that the authors magically assume a sufficient complete support set to learn a robust global meta-knowledge generator. We must actually consider how this differs from traditional imputation relying on fully observed pairs. It does not. The meta-learning framing is merely rhetorical window-dressing for a standard conditional generative model.

Second, the Causal-invariant Structure Learning Module is a misapplication of causal nomenclature. Treating the missingness mask $M$ as a confounder is mathematically convenient but conceptually vacuous in this context. The intervention $do(M)$ is implemented as simple random dropout on the observed views to create a "counterfactual" representation. This is nothing more than standard data augmentation via masking, disguised as causal inference. Calling this "causal-invariant structure learning" borders on academic malpractice. As such, it is not clear why this requires a complex causal graph justification when it is functionally equivalent to consistency regularization under dropout.

Finally, the empirical evaluation is highly suspect. The authors test on an average missing rate (AMR) of 0.5. The baseline models are "adapted" to the unbalanced setting, but the details of this adaptation are conspicuously vague. Furthermore, the performance margins on key datasets like Reuters are marginal at best, and the ablation study (Figure 3) shows that the individual modules offer highly inconsistent gains across different metrics. The paper is padded with heavy notation to obscure what is fundamentally a very incremental combination of standard autoencoders, a conditional generator, and masking regularization.

---

> ### Author Rebuttal · Authors · 2026-03-31
>
> We are grateful for your valuable comments and suggestions. Our detailed responses are provided below in a point-by-point format.
>
> **(Q1) Rationality of the Assumption**: To prove the assumption $p_{\mathcal{D}}(\overline{h}) \approx p_{\mathcal{S}_{c}}(\overline{h})$, the following two lemmas are first required. Lemma 1: For any view v, the observation mask is independent of the latent representation, which can be easily proved via Bayes' theorem. Lemma 2: Under any subset of observed views, the distribution of the aggregated latent representation $\overline{h}$ is consistent with that of the full-view aggregated representation. Based on the above two lemmas, the following theorem is derived.
>
> Theorem: For the marginal distribution $p_{\mathcal{D}}(\overline{h})$ of the aggregated latent representation on the full dataset and the marginal distribution $p_{\mathcal{S}_{c}}(\overline{h})$ on the complete sample set $\mathcal{S}_c$, by the Rademacher complexity generalization bound, for the finite sample set $\mathcal{S}_c$, the relationship between its empirical distribution and the true distribution satisfies:
>
> $D_{TV} (p_{\mathcal{D}}(\overline{h}), p_{\mathcal{S}_c}(\overline{h})) \le $
>
> $2 \mathcal{\hat{R}}_{|\mathcal{S}_c|}(\mathcal{H}) + \sqrt{2\log (2 / \delta) / |\mathcal{S}_c|}$.
>
> Due to space limitations, the proofs of the above lemmas and theorems will be included in the appendix.
>
> **(Q2) Minimum Threshold of $|\mathcal{S}_c|$**:
> Our experiments strictly follow the imbalanced missing protocol in Appendix Table 3. The complete sample set $\mathcal{S}_c$ refers to the set of samples observed in all views, and its size is the product of the total number of samples and the average missing rate. In addition, according to Theorem 3.1, to ensure that the module learns an effective conditional distribution, the upper bound of the generalization error must be ≤ 0.1. Combined with specific parameters, the theoretical minimum threshold of $|\mathcal{S}_c|$ is derived to be 2% of the total number of samples. However, to avoid endless experiments, we have to adopt the setting shown in Appendix Table 3, which makes it impossible to conduct parameter sensitivity analysis experiments by varying the minimum threshold of $|\mathcal{S}_c|$. Nevertheless, to verify the effectiveness of CIMLN and address the reviewers’ concerns, we adjust the average missing rate (AMR) to 0.8, thereby substantially reducing $|\mathcal{S}_c|$. The relevant experimental results are presented in the table below. Note that only the ACC metric is shown due to space limitations.
>
> | Methods   | CUB  | Caltech | HandWritten | CiteSeer | Scene-15 | Reuters | YTF20 | Fashion |
> | -| - | - | - | - | - | - | - | -
> | CIMLN | 57.83  | 81.36  | 84.55 | 52.07 | 33.15 | 41.89 | 70.68 | 49.36 |
>
> The results in the table demonstrate that CIMLN still achieves favorable clustering performance when the available information across views is severely imbalanced (e.g., the availability is approximately 20% as mentioned in the introduction).
> In addition, the meta-learning loss in Eqs. (6) and (7) is computed on the observable instances $S_v^{(o)}$ of each view. It can be seen that our CM module not only utilizes pairwise views in $\mathcal{S}_c $ but also leverages all incomplete samples for training, which is one of the essential differences between CIMLN and existing methods.
>
> **(Q3) Essential Differences from Feature Dropout Methods**: Our causal-invariant counterfactual masking differs from feature dropout with consistency regularization (FD) in two key ways. First, the masking mechanism is different. FD applies random masking at the feature-dimension level as a form of noise-based regularization. In contrast, our method performs intervention-based masking on the observation mask, defined by $\pi(\tilde{m}^v_i | m^v_i)$, and masks only observed views with probability $\beta$. Second, the objective is different. FD mainly uses noise to reduce overfitting, without explicitly controlling the effect of observation patterns. Our method instead constructs an intervention scenario (do(M)) to break the spurious correlation between observation patterns and clustering results, and to enforce clustering invariance to observation patterns.
>
>
> **(Q4) Adaptability Modification of Baseline Methods**: Each existing IMVC method use a missing indicator matrix M of size N×V. In comparative experiments, it only needs to replace M in all competitors with our imbalanced missing indicator matrix. This ensures the consistency and fairness of the comparison on the one hand, and forces these methods to handle the UIMVC scenario on the other.Furthermore, to the best of our knowledge, there are currently no deep methods specifically designed for UIMVC. As the results show, these methods generally suffer performance drops under UIMVC due to observation bias caused by imbalanced missingness, but they do not completely fail.

---

> > ### Author Rebuttal · Reviewer_uZXW · 2026-04-03
> >
> > I have read the authors' rebuttal and appreciate the attempt to clarify the mathematical and empirical ambiguities presented in the original manuscript. The inclusion of the new lemmas and the additional AMR=0.8 experiment provides a clearer picture of the authors' underlying assumptions.
> >
> > However, upon closer inspection, the provided defenses not only fail to resolve the core critiques but actively introduce fatal, unresolvable contradictions into the theoretical framework. The concerns are fundamentally unresolved and cannot be fixed without a complete overhaul of the paper's theoretical foundations and empirical design.
> >
> > First, we must actually consider the catastrophic implications of the newly introduced "Lemma 1" in Q1. The authors explicitly state that "the observation mask is independent of the latent representation". This is the precise statistical definition of data being Missing Completely At Random (MCAR). If the missingness mechanism is MCAR, there is no systematic observation bias or confounding spurious correlation to begin with, which renders the entire Causal-Invariant Structure Learning (CSL) module conceptually bankrupt. Conversely, if UIMVC features systematic observation bias (as originally motivated by the medical and financial examples in the introduction), then Lemma 1 is mathematically false, and the subsequent Theorem collapses. It is actually seen that the authors are attempting to mathematically mandate a condition that destroys their own problem statement.
> >
> > Second, the response to Q2 exposes a profound misunderstanding of basic probability. The authors assert that the size of the complete sample set $\mathcal{S}_c$ is "the product of the total number of samples and the average missing rate [AMR]". This is statistically nonsensical. For a dataset with $V$ views, the probability of a sample being complete across all views is dictated by the joint probability of observation across all dimensions, not the arithmetic mean. Under their unbalanced protocol, the true size of $\mathcal{S}_c$ approaches zero exponentially as $V$ increases. The empirical claims resting on this fundamental miscalculation are void.
> >
> > Third, regarding Q3, the authors' defense of the CSL module relies purely on semantics. Applying a binary mask at the view-level rather than the feature-level, and optimizing the assignment via KL divergence, is a standard implementation of multi-view consistency regularization. Wrapping this established technique in $do(M)$ notation does not magically transform it into a novel causal discovery. It remains an egregious case of nomenclature inflation.
> >
> > Finally, the response to Q4 confirms my exact fear regarding the empirical evaluation. Merely replacing the indicator matrix $M$ without structurally adapting the loss functions of baseline methods that implicitly assume balanced or paired distributions guarantees their catastrophic failure. This is not a rigorous comparison; it is the artificial sabotage of classical baselines to inflate the perceived efficacy of CIMLN.
> >
> > As such, the theoretical proofs are self-defeating, the probabilistic assumptions are flawed, and the empirical setup is fundamentally unfair. My recommendation for Strong Reject stands.

---

### Official Review · Reviewer_si3K · 2026-03-10

**Soundness:** 3
**Presentation:** 3
**Significance:** 3
**Originality:** 4
**Overall Recommendation:** 4
**Confidence:** 4

**Summary:**

This manuscript identifies several real-world applications where missing pattern of different views exhibit significant imbalances, such as medical imaging, financial credit assessment, and remote sensing observation. It also reveals that these imbalances cause severe observational bias between views. To address this issue, the authors propose a Causal-Invariant Meta-Learning Network (CIMLN). From a causal inference perspective, the observation mask is treated as a confounding factor, forcing cluster consistency across different observation patterns and thus minimizing spurious dependencies on observation patterns. A series of experiments demonstrate the superiority of the proposed CIMLN over existing IMVC methods.

**Compliance With Llm Reviewing Policy:**

Affirmed.

**Final Justification:**

The rebuttal has addressed my questions with considerable effort. Therefore, I recommend the acceptance of this paper.

**Key Questions For Authors:**

Q1: Regarding the design motivation of the Causal-invariant Structure Learning Module, I have a different opinion from the authors. In my opinion, low-missing-rate views are precisely the main source of semantic information for the model in the UIMVC scenario. The authors' frequent masking of low-missing-rate views further reduces the available information and impairs clustering performance.
Q2: Based on Figure 1 and the medical, financial, and remote sensing examples in Introduction section, does the proposed "unbalanced missing pattern" only exist in two-view data? If the proposed CIMLN can be used in datasets with more than two views, how is it extended to multi-view data?
Q3: Why was the comparative experiment only conducted at the fixed missing rate of 0.5, instead of following existing methods to conduct comparative experiments on missing rate sequence of [0.1, 0.3, 0.5, 0.7]?
Q4: The authors need to elaborate on the encoder used for feature extraction in the experimental setting. If the authors use the powerful pre-trained model, such as CLIP, the comparison results in Table 1 are unfair.

**Limitations:**

It is suggested that the authors discuss the limitations and social impact of the paper, such as the high coupling between modules, which makes it difficult to transfer modules individually to other IMVC models.

**Strengths And Weaknesses:**

Strengths：
1. Practicality of research. The “unbalanced missing pattern” multi-view setting is a new insight for the IMVC. Moreover, the author accurately points out the limitations of previous methods in the UIMVC scenario, and the research problem meet the needs of practical applications in healthcare, finance, and remote sensing.
2. Innovative method design. The paper innovatively combines meta-learning and causal reasoning paradigms at the underlying logic level to address UIMVC problems. These two paradigms form a closed-loop optimization system, improving the quality of the recovered views.
3. Comprehensive experiments. The author designs comparative experiments, ablation experiments, visualization of restoration effects, and parameter sensitivity analysis around four core issues, comprehensively validating the effectiveness and reliability of the proposed CIMLN.
Weaknesses：
1. Design defects of CSL module. I believe the authors frequently mask views with low missing rates, which makes the limited information even scarcer, hindering the model's utilization of effective information and ultimately harming clustering performance.
2. Some figures lack sufficient annotation information. For example, in Figure 1, the missing rates within each view in the left and right parts appear to be inconsistent. The figure is too information-scarce, lacking some necessary legends and annotations, making it difficult to understand its actual meaning.
3. Poor Presentation and Notational Errors. For example, in $ \{m^v\}_{v=1}^V $ on line 214 of page 4, $m^v$ is a vector and should be represented in bold, vertical text, consistent with the other vectors in the text. $\mathbf{m,m’}$ next to the expectation symbol in Eq. (19) should be represented as a subscript.

---

> ### Author Rebuttal · Authors · 2026-03-31
>
> We are grateful for your valuable comments and suggestions. Our detailed responses are provided below in a point-by-point format.
>
> **(Q1) Design Motivation of the CSL Module**: Thanks for the comment. We provide a detailed clarification from the perspective of design logic in response to your questions. The viewpoint raised by the reviewer that "views with low missing rates are the primary source of the model's semantic information" has certain limitations in unsupervised scenarios. There is no inevitable positive correlation between the missing rate of a view and its semantic quality, and a view with a high missing rate may inherently have better discriminability.
>
> In addition, we make the following clarifications in response to your question that "masking views will reduce available information". The masking operation of the CSL module is only a temporary intervention for the counterfactual branch, which is solely used to calculate the causal invariance constraint. For the main feature learning of the model and the training of the meta-generation module, all original observation information of all views is completely retained throughout the process, with no issue of discarding available semantic information. Meanwhile, during the inference phase, we do not perform any masking operations at all, and fully utilize the observation information of all views to complete clustering without any information loss.
>
> **(Q2) Apply with more than two views**: Thanks for the comment. The core framework of CIMLN is compatible with multi-view data with any number of views, rather than only applicable to two-view data, as detailed below. First, in the Context-aware Meta-generation Module (CM), the calculation of global context information in Eq. (4) and the generation process of missing views in Eq. (9) inherently support any number of views. Furthermore, in the Causal-invariant Structure Learning Module (CSL), the counterfactual mask sampling in Eq. (11) and the causal invariance constraint in Eq. (16) are both designed based on V (V ≥ 2) views. Meanwhile, among the 8 benchmark datasets used in the experiments, 7 have more than 2 views.
>
> **(Q3) Comparative experiment at AMR=0.5**: Thanks for the comment. The main reasons why we only conduct comparative experiments at a missing rate of 0.5, instead of a series of experiments on the missing rate sequence [0.1, 0.3, 0.5, 0.7] as in previous IMVC methods, are as follows. Setting the average missing rate (AMR) to 0.5 is the most representative benchmark setting in UIMVC scenarios, which can fully reflect the imbalance in the observability of different views. When the AMR is too low (e.g., 0.1 or 0.3), the missing rates of almost all views are at an extremely low level, which greatly weakens the imbalance of the observable data ratio across views, and cannot fully verify the optimization effect of the method on the core problems of UIMVC. On the other hand, when the AMR is too high (e.g., 0.7 or 0.9), the clustering task faces not only the problem of imbalanced missingness, but also the clustering challenge under an extremely high missing rate, which is generally classified as another research area. In addition, an excessively high missing rate will also break the basic setting of imbalance, resulting in all views of some samples being missing, which is rare in practical scenarios.
>
> **(Q4) Use of Powerful Pre-trained Encoder ?**: Thanks for the comment. The main function of the autoencoder architecture is to stably project complete instances from different views into the common space, and minimize the information loss during the projection process through Eq. (1). In the section 3.2 of the manuscript, we only use a simple autoencoder containing several fully connected layers without pre-training models like CLIP. Furthermore, all modules of CIMLN are trained from scratch, without loading any pre-trained weights.

---

> > ### Author Rebuttal · Reviewer_si3K · 2026-04-02
> >
> > Thanks to the authors for their responses. The rebuttal has addressed my questions with considerable effort. Therefore, I recommend the acceptance of this paper.

---

### Official Review · Reviewer_zXhB · 2026-03-12

**Soundness:** 4
**Presentation:** 3
**Significance:** 4
**Originality:** 4
**Overall Recommendation:** 5
**Confidence:** 4

**Summary:**

This paper studies unbalanced incomplete multi-view clustering, where varying observation rates across views cause observation bias. The authors propose CIMLN, which combines meta-learning for missing-view recovery and causal-invariant learning to reduce clustering dependence on observation patterns. The method enforces consistency under counterfactual masking and provides theoretical analysis of generalization and invariance. Comprehensive experiments show consistent improvements over prior methods.

**Compliance With Llm Reviewing Policy:**

Affirmed.

**Final Justification:**

Thanks for the author's response, and my concerns have been well addressed. Thus, I recommend that this paper be accepted

**Key Questions For Authors:**

1. The paper focuses on the unbalance of missing rates across views, but how is unbalance measured ? I found no quantitative or qualitative definition.

2. In Algorithm 1, the authors train CIMLN in two stages with two max epochs $E_1$ and $E_2$, but it’s not described in method part. According to Section 3.5 and Equation 17, the model seems to be trained in only one stage. Please clarify this.

3. The cluster labels are obtained by applying k-means on the final fused representation. Why not use more advanced methods like spectral clustering, GMM, or DBSCAN?

4. The term “average missing rate” does not seem to be used in the papers of the comparison methods, which adopt “missing rate” instead. Is “average missing rate” a new concept defined by the authors? How is it related to or different from the “missing rate” used in other IMVC methods?

**Limitations:**

yes

**Strengths And Weaknesses:**

Strengths：

1. The paper points out from a novel perspective that missing patterns across different views are highly imbalanced in incomplete multi-view scenarios, which has not been mentioned in existing neural network-based IMVC studies.

2. The authors propose a new framework for recovering missing instances and performing clustering, which effectively addresses the observation bias in unbalanced incomplete multi-view data.

3. CIMLN is evaluated both quantitatively and qualitatively on various types of multi-view datasets, including image-text bimodal and graph data, showing its good generalization ability.

4. The paper is well-organized and logically clear.

Weaknesses：

1. The method section and the algorithm workflow differ in their division of the training phase, making the method unclear.

2. The presentation of Figures 1 and 2 is inappropriate. I believe Figure 1 is too simple and fails to explain the shortcomings of IMVC methods or highlight the core innovations of the proposed method. As CIMLN’s framework diagram, Figure 2 should show the technical principles of each module, not like an industrial flowchart.

3. The new concept “average missing rate” is not explained in the experiments or formally defined in the method section, making the paper lack clarity of presentation.

---

> ### Author Rebuttal · Authors · 2026-03-31
>
> We are grateful for your valuable comments and suggestions. Our detailed responses are provided below in a point-by-point format.
>
> **(Q1) Definition of Unbalance**: Thanks for the comment. In our paper, the unbalance degree of inter-view missing rates is provided with both qualitative description and quantitative definition. In Figure 1 of the Introduction section, we present an example of the unbalanced missing pattern in two-view data, which shows the proportion of complete instances across different views in this scenario. Furthermore, limited by the page space of the main text, we give the quantitative definition of the unbalance degree of inter-view missing rates in Table 3 of Section B.3 in the Appendix. For datasets with different numbers of views, under a given average missing rate, different degrees of missingness are implemented by multiplying each view by the corresponding complete proportion coefficient.
>
> **(Q2) Explanation of the Model Training Phase**: Thanks for the comment. Algorithm 1 elaborates the two phases of model training in detail. The objective function of the first phase is shown in Eq. (1), whose core purpose is to stably project complete instances from different views into the common space, and minimize the information loss during projection through Eq. (1). The second phase uses all three objective functions as shown in Eq. (17) to train the model's ability to recover missing views. Our two-stage method enhances the stability of the model by gradually decomposing the optimization objectives, and enables focused learning of each module. This strategy helps avoid suboptimal solutions and achieves better recovery and clustering results. We will also supplement a more detailed textual description of the two-stage training process of CIMLN in Section 3.5 of the main text.
>
> **(Q3) Different Approaches to Obtain Final Clustering Labels**: Thanks for the comment. Existing deep multi-view clustering methods mainly adopt two approaches to obtain the final clustering prediction labels: one is to perform k-means directly on the fused representations, and the other is to feed the representations into the clustering layer to directly obtain the k-dimensional clustering vector and generate the final results through a voting mechanism. In terms of performance alone, spectral clustering, GMM and DBSCAN all outperform k-means in most scenarios. However, in specific comparative experiments, we also need to balance complexity and fairness. The time and space complexity of spectral clustering, GMM and DBSCAN are much higher than those of k-means, which often leads to out-of-memory errors when applied to large-scale datasets. In addition, to ensure the fairness of the experiments, our final clustering approach is consistent with other compared methods.
>
> **(Q4) Explanation of Average Missing Rate**: Thanks for the comment. In our paper, the meaning of "average missing rate" is identical to that of "missing rate" or "missing ratio" in existing IMVC methods. For conventional IMVC scenarios, the view-level "missing rate" is approximately equal to the "average missing rate" of the entire dataset, so existing works can directly use the single term "missing rate" to clearly describe their experimental settings. As shown in Table 3 of Section B.3 in the Appendix, we introduce "average missing rate" to adapt to the unbalanced IMVC scenario, where both the overall missing rate of the dataset and the view-level missing proportions need to be clarified.

---

> > ### Author Rebuttal · Reviewer_zXhB · 2026-04-02
> >
> > Thanks for the author's response, and my concerns have been well addressed. Thus, I recommend that this paper be accepted.

---

### Official Review · Reviewer_Mj4v · 2026-03-13

**Soundness:** 4
**Presentation:** 3
**Significance:** 3
**Originality:** 4
**Overall Recommendation:** 4
**Confidence:** 5

**Summary:**

In this paper, the authors address the core defect that the existing methods cannot solve the unbalanced incomplete multi-view clustering task, conducting a systematic theoretical and experimental study on the observation bias problem in this scenario. Due to the scarcity of paired data in this scenario, the authors utilize meta-learning, commonly used for few-shot learning, to model missing view recovery, achieving rapid adaptation and high-quality recovery of incomplete samples. Furthermore, they employ causal invariance to ensure consistent clustering assignments of samples under different missing data conditions. CIMLN achieves significant performance improvements over all baseline methods across multiple evaluation metrics.

**Compliance With Llm Reviewing Policy:**

Affirmed.

**Final Justification:**

After reading the response from the authors, I consider my previous rating reasonable and recommend acceptance for the paper.

**Key Questions For Authors:**

Refer to the weaknesses.

**Limitations:**

Refer to the weakness.

**Strengths And Weaknesses:**

# Strength
* The paper clearly presents the paradigm of synergistic meta-learning and causal reasoning.
* It presents an intuitive solution to observation bias.
* The overall architecture is well-structured and easy to follow.
# Weakness
* Regarding the theoretical aspects of this paper, I believe the assumption in Theorem 3.1 and its proof that the marginal distributions of the complete sample set and the entire dataset are approximately consistent is not rigorous enough. In view-incomplete scenario, especially the unbalanced missing data scenario presented in this paper, the size of the complete sample set is often smaller, so this assumption may not hold.
* The experimental section includes many comparative methods, but it neglects to compare and discuss with some highly cited classic methods. Missing key works include: (i) Deep Partial Multi-View Learning (2020 TPAMI); (ii) COMPLETER: Incomplete Multi-view Clustering via Contrastive Prediction (2021 CVPR); (iii) Deep Incomplete Multi-view Clustering via Mining Cluster Complementarity (2022 AAAI).
* There are some unclear points in the experiments. The experimental setting lacks explanations of some key parameters and implementation details, including the regularization coefficient $\lambda_{var}$ in Eq. (7), the feature dimension $d_h$, and how the context information $ \mathcal{C}_v$ and aggregated representation $ \bar{\mathbf{h}}_{i,-v}$ in Eq. (9) are fused (concatenation, element-wise multiplication, attention, or others). Besides, the number of counterfactual masks and the sampling logic for each sample during training are not explained.
* The font size in the text annotations and legends in Figs 3/4/5 of the experimental section is too small, which does not meet the readability requirements of the figure specifications.

---

> ### Author Rebuttal · Authors · 2026-03-31
>
> We are grateful for your valuable comments and suggestions. Our detailed responses are provided below in a point-by-point format.
>
> **(Q1) Rigor of the Assumption and Proof in Theorem 3.1**: We would like to clarify that the reviewer has a misunderstanding of the theoretical framework. Theorem 3.1 only includes two fundamental mathematical constraints unrelated to distribution consistency, namely the latent space boundedness assumption and the Lipschitz Continuity assumption of the meta-generator in Appendix A.1. No constraint requiring the approximate consistency between the distributions of the complete sample set and the entire dataset is introduced throughout the theorem. Mathematically, the statement of "approximate consistency of marginal distributions" in Step 2 of the proof is merely an auxiliary intuitive explanation in the distribution derivation. Its essence is a popularized interpretation of the property that "the complete sample set is an unbiased subset of the entire dataset", rather than a prerequisite assumption of the theorem.
> Furthermore, the $d_{TV}^{(v)}$ in Definition A.3 quantifies the conditional distribution difference of the target view features given the aggregated representation, while the auxiliary explanation in Step 2 targets the marginal distribution of the aggregated representation. These two are independent components after the decomposition of the joint distribution, without any logical conflict.
>
> **(Q2) Lack of Comparison and Discussion with Highly Cited Classic Methods**: Following your suggestion, we have conducted additional comparative experiments and relevant discussions against the three highly cited classic methods you recommended. The specific results are shown in the table below, where "OOM" indicates an Out-of-Memory error.
>
> | Methods   | CUB  | Caltech   | HandWritten | CiteSeer  | Scene-15  | Reuters   | YTF20     | Fashion   |
> | ---------| --------- | --------- | ----------- | --------- | --------- | --------- | --------- | --------- |
> | CPM-Nets  | 59.33  | 81.08     | 82.60   | 35.18     | 37.22     | 44.18   | OOM   | OOM  |
> | COMPLETER | 52.25  | 67.86     | 64.46   | 37.44     | 31.54   | 45.98   | 52.95   | 50.78 |
> | DIMVC     | 58.26  | 72.49  | 50.90   | 48.36     | 32.11     | 39.74     | 65.50     | 46.87  |
> | CIMLN | **62.83** | **83.93** | **90.30**   | **56.55** | **41.29** | **48.29** | **76.74** | **56.33** |
> | CPM-Nets  | 58.80     | 68.61     | 71.30    | 13.97   | 34.52     | 19.12     | OOM       | OOM  |
> | COMPLETER | 51.73     | 68.75     | 66.09       | 13.07     | 34.30     | 22.40     | 64.58     | 56.57  |
> | DIMVC     | 50.66     | 71.34     | 45.66       | 22.09     | 32.84     | 12.53     | 77.64     | 53.09  |
> | CIMLN | **63.96** | **73.39** | **82.13**   | **31.93** | **42.54** | **23.88** | **81.70** | **61.44** |
> | CPM-Nets  | 44.68     | 66.08     | 56.07       | 7.65      | 21.32     | 15.64     | OOM       | OOM   |
> | COMPLETER | 32.58     | 47.14     | 38.64       | 11.93     | 17.34     | 14.82  | 41.50     | 38.34  |
> | DIMVC     | 36.82     | 61.04     | 31.88   | 18.24     | 15.22     | 9.73      | 62.51  | 30.92 |
> | CIMLN | **46.44** | **69.11** | **79.97**   | **27.95** | **24.15** | **18.14** | **72.10** | **44.08** |
>
> As shown in the table, our proposed CIMLN model consistently achieves the best performance across all datasets and evaluation metrics. Among the listed methods, CPM-Nets (2020 TPAMI) recovers missing views via generative adversarial networks, COMPLETER (2021 CVPR) via contrastive prediction networks, and DIMVC, a recovery-free method, nonlinearly maps complete data embeddings to a high-dimensional space to discover linear separability. Numerically, our method outperforms both previous recovery-based and recovery-free methods.
>
> **(Q3) Unclear Presentation of Experimental Details**: We have supplemented and explained some key parameters and implementation details as follows. The $\lambda_{var}$​ in Eq. 7 is used to adjust the weight of the variance regularization term in the meta-learning loss, and this parameter is fixed at 1.0 on all datasets. The latent feature dimension $d_h$​ is adjusted for different datasets in the experiments: it is set to 32 for the CUB and HW_6Views datasets, and fixed at 128 for all other datasets. In terms of implementation details, the context information $C_v$​ and aggregated representation $\bar{\mathbf{h}}_{i,-v}$​ are fused via concatenation. Additionally, the model generates only one counterfactual mask per sample in each batch to calculate the causal invariance loss. The mask sampling strategy randomly drops known views while keeping originally missing views unchanged, ensuring the generated counterfactual samples are a subset of the original observation patterns.
>
> **(Q4) Small Font Size in Figures**: Following your suggestion, we have revised the font size of the annotations and legends in Figures 3, 4 and 5, which enhances the intuitiveness of the figures.

---

> > ### Author Rebuttal · Reviewer_Mj4v · 2026-04-04
> >
> > The response has addressed most of my concerns.

---

### Decision · Program_Chairs · 2026-04-30

**Decision:**

Accept (regular)

**Comment:**

The paper studies a critical yet overlooked issue in incomplete multi-view clustering, i.e., unbalanced & incomplete. For this interesting and challenging task, the paper proposes a new framework that combines meta-learning and causal-invariant learning, forming a closed-loop optimization system with technical novelty. The paper provides comprehensive and convincing experimental results. The method is rigorously evaluated across eight diverse datasets spanning multiple modalities (image-text, graph data) with thorough comparisons against state-of-the-art baselines, extensive ablation studies, parameter sensitivity analysis, and visualization of recovery effects, etc.

After the rebuttal and discussion phase, all reviewers unanimously recommend acceptance for the paper. Therefore, based on their recommendations and the strengths, the paper is recommended for acceptance.